# Malignant Melanoma: An Overview, New Perspectives, and Vitamin D Signaling

**DOI:** 10.3390/cancers16122262

**Published:** 2024-06-18

**Authors:** Radomir M. Slominski, Tae-Kang Kim, Zorica Janjetovic, Anna A. Brożyna, Ewa Podgorska, Katie M. Dixon, Rebecca S. Mason, Robert C. Tuckey, Rahul Sharma, David K. Crossman, Craig Elmets, Chander Raman, Anton M. Jetten, Arup K. Indra, Andrzej T. Slominski

**Affiliations:** 1Department of Rheumatology and Clinical Immunology, Department of Medicine, School of Medicine, University of Alabama at Birmingham, Birmingham, AL 35294, USA; rslominski@uabmc.edu; 2Department of Dermatology, School of Medicine, University of Alabama at Birmingham, Birmingham, AL 35294, USA; tkim@uabmc.edu (T.-K.K.); zjanjetovic@uabmc.edu (Z.J.); epodgorska@uabmc.edu (E.P.); celmets@uabmc.edu (C.E.); craman@uabmc.edu (C.R.); 3Department of Human Biology, Institute of Biology, Faculty of Biological and Veterinary Sciences, Nicolaus Copernicus University, 87-100 Torun, Poland; anna.brozyna@umk.pl; 4School of Medical Sciences, The University of Sydney, Sydney, NSW 2050, Australia; katie.dixon@sydney.edu.au (K.M.D.); rebecca.mason@sydney.edu.au (R.S.M.); 5School of Molecular Sciences, University of Western Australia, Perth, WA 6009, Australia; robert.tuckey@uwa.edu.au; 6Department of Biomedical Informatics and Data Science, School of Medicine, University of Alabama at Birmingham, Birmingham, AL 35294, USA; rsharma3@uab.edu; 7Department of Genetics and Bioinformatics, School of Medicine, University of Alabama at Birmingham, Birmingham, AL 35294, USA; dcrossman@uabmc.edu; 8Cell Biology Section, NIEHS—National Institutes of Health, Research Triangle Park, NC 27709, USA; jetten@niehs.nih.gov; 9Department of Pharmaceutical Sciences, College of Pharmacy, Oregon State University, Corvallis, OR 97331, USA; 10Department of Dermatology, Oregon Health & Science University, Portland, OR 97239, USA; 11Knight Cancer Institute, Oregon Health & Science University, Portland, OR 97239, USA; 12Comprehensive Cancer Center, University of Alabama at Birmingham, Birmingham, AL 35294, USA; 13Pathology and Laboratory Medicine Service, Veteran Administration Medical Center, Birmingham, AL 35233, USA

**Keywords:** melanoma, melanomagenesis, vitamin D, nuclear receptors, prevention, therapy, tumor progression, bioinformatics

## Abstract

**Simple Summary:**

Despite recent advances in diagnosis and therapy, malignant melanoma poses a significant problem both to clinicians and cancer researchers due to its resistance to therapy and unpredictable behavior. In this review, we discuss etiology, risk factors, diagnosis, prognosis, and therapy of melanoma, with a focus on new developments in these areas including bioinformatics. These are analyzed in the context of its unique metabolic characteristics and of recent advances in vitamin D biology with implications for melanoma. Active forms of vitamin D can prevent or inhibit melanoma development and progression, and can be used in therapy of this disease. Knowledge of patient vitamin D status and vitamin D signaling in the tumoral tissue can help in predicting the progression of the disease and in primary or adjuvant therapy. Therefore, vitamin D signaling represents a realistic target for the prevention or therapy of malignant melanoma.

**Abstract:**

Melanoma, originating through malignant transformation of melanin-producing melanocytes, is a formidable malignancy, characterized by local invasiveness, recurrence, early metastasis, resistance to therapy, and a high mortality rate. This review discusses etiologic and risk factors for melanoma, diagnostic and prognostic tools, including recent advances in molecular biology, omics, and bioinformatics, and provides an overview of its therapy. Since the incidence of melanoma is rising and mortality remains unacceptably high, we discuss its inherent properties, including melanogenesis, that make this disease resilient to treatment and propose to use AI to solve the above complex and multidimensional problems. We provide an overview on vitamin D and its anticancerogenic properties, and report recent advances in this field that can provide solutions for the prevention and/or therapy of melanoma. Experimental papers and clinicopathological studies on the role of vitamin D status and signaling pathways initiated by its active metabolites in melanoma prognosis and therapy are reviewed. We conclude that vitamin D signaling, defined by specific nuclear receptors and selective activation by specific vitamin D hydroxyderivatives, can provide a benefit for new or existing therapeutic approaches. We propose to target vitamin D signaling with the use of computational biology and AI tools to provide a solution to the melanoma problem.

## 1. Introduction

Malignant melanomas result from the malignant transformation of melanocytes, melanin pigment-producing cells of neural crest origin [1,2,3]. The melanogenic activity and behavior of melanocytes are regulated by numerous factors, such as ultraviolet radiation (UVR) and chemical and biological mediators encompassing hormonal and non-hormonal regulators, as well as genetic and molecular ones [1,2,3,4,5,6,7,8,9,10,11,12,13,14]. The most prevalent types of melanomas include cutaneous malignant melanomas which affect large segments of the population with high incidence and mortality rates compared to other cancers [2,6,11,15,16,17,18,19]. However, melanomas also originate from melanocytes of the eye and mucosa (including oral, anorectal, and genitourinary melanomas) with very rare melanomas arising from other organs including the central nervous system (CNS) [20,21,22,23,24,25,26,27,28,29]. According to the recent World Health Organization (WHO) classification, melanomas are subdivided etiologically into those related to sun exposure and those that are not, with the former stratified according to cumulative solar damage (CSD) of the skin into low- and high-CSD melanomas [2,21,30]. Melanomas of which the etiology is predominantly unrelated to sun exposure include mucosal, acral, uveal and Spitzoid melanomas, melanomas arising in blue or congenital nevi, and rare melanomas arising in the CNS [7,21,25,27,31,32,33,34,35]. In this review, we will focus on melanomas related to sun exposure, including melanomas with low CSD, such as superficial spreading type and nodular melanomas, and with high CSD, such as the lentigo maligna type and desmoplastic melanomas [21,30,36,37].

Cutaneous melanoma represents a significant clinical problem due to its recalcitrance. It encompassed 3–6% of all new cancer cases in the USA in 2016, excluding basal and squamous cell carcinomas (BCC and SCC) [16]. The most efficient methods of melanoma management involve prevention, screening, early diagnosis, and surgical excision when the disease is localized to the skin [9,30,38,39]. Recently, there have been significant advancements in therapies for treating stage III and IV melanomas, including targeting molecular pathways using BRAF and MEK inhibitors or immune checkpoint inhibitors, such as PD-1, CTLA-4, or LAG-3 inhibitors [6,18,30,36,40,41,42,43,44,45,46,47,48]. The BRAF and MEK inhibitors have shown impressive clinical responses [47]. However, the duration of efficacy only lasts a few months before resistance develops and the patients die. The utility of the above strategies is somewhat limited because of adverse effects, financial costs, and inherent or acquired tumor resistance mechanisms leading to recurrent disease and death of the patient [44,49,50]. Moreover, we do not have a rational strategy to treat patients at high risk for metastatic disease without a pathological proof of metastases. Thus, defining new regulatory targets such as nuclear receptors including the vitamin D receptor (VDR) [14,51,52,53,54,55,56,57,58] and natural compounds such as vitamin D3 [59,60,61,62] and its biologically active hydroxyderivatives that are economical with limited or no side effects [63,64,65,66] is needed. In this paper we will discuss traditional [67,68,69,70,71] and novel vitamin D signaling pathways [64,72,73] as targets for melanoma prevention and therapy, in a context dependent fashion. Bioinformatics, systems biology, and personalized medicine approaches will also be utilized in this task to integrate genomic and non-genomic actions of vitamin D, its hydroxyderivatives with their cognate receptor(s), and other signaling pathways involved.

## 2. Cutaneous Melanoma in a “Nutshell”

### 2.1. Etiology

Malignant transformation of melanocytes with further progression to advanced stages, collectively called melanomagenesis, is initiated and driven by environmental, genetic (inheritable), constitutional, and epigenetic factors, as well as by acquired mutations with the accumulation of genomic changes further amplified by local and systemic neuroimmunoendocrine factors affecting progression of the disease [1,2,4,6,9,11,12,13,14,30,32,50,74,75,76,77,78,79,80,81,82,83,84,85,86,87,88]. Since these factors have been discussed in many review articles, we will only provide a brief overview of them.

Approximately 10% of melanoma cases are linked to inherited genetic mutations and meet the criteria of familial melanoma [1,2,6,9,13,14,38,40,44,89,90,91,92,93,94,95,96,97,98]. Thus, mutations in certain loci are linked with increased risk of melanoma. These include high (*CDKN2A*, *CDK4*, *BAP1*, *TERT*, *TERF2IP*, *POT1*, *POLE*, and *ACD*), medium (*MITF*, *MC1R,* and *SLC45A2*), and relatively lower (*Mdm2*, *RB1*, *ASIP*, *TYR*, *TRP1*, *OCA2*, and *VDR*) penetrance genes [2,13,44,91,99,100]. The classical and considered to be the most important gene, accounting for 20–40% of predisposition for melanoma, is *CDKN2A.* It encodes two tumor suppressor proteins, p16 (INK4A) and p14 (ARF), that inhibit the cell cycle and enhance apoptosis through action on *CDK4* [encoding cyclin-dependent protein kinase 4 (CDK4)] and inhibition of CDK6 kinases [13,30,99,101,102]. Patients carrying this mutation predominantly develop superficial spreading melanoma, with much lower frequency for acral or nodular melanoma [99,103]. High penetrance genes involved in telomere maintenance are *TERT*, *TERF2IP*, *POT1*, *POLE*, and *ACD*, encoding the following proteins, respectively: telomerase reverse transcriptase, telomeric repeat binding factor 2 interacting protein, protection of telomeres 1, DNA polymerase epsilon catalytic subunit, and adrenocortical dysplasia protein homolog [2,13,44,91,99]. BRCA1-associated protein 1 (*BAP1*) is a tumor suppressor gene in which germline inactivating mutations lead to atypical intradermal melanocytic tumors and a small proportion of sporadic cutaneous melanoma [91]. The medium penetrance genes, in which mutations are linked to melanoma, are *MITF*, *MC1R*, and *SLC45A2*, and encode microphthalmia-associated transcription factor (master regulator of melanocytic activity) melanocortin receptor type 1 (G-protein coupled membrane bound receptor) and solute carrier family 45, member 2 (regulates melanosomal pH and trafficking of melanogenesis-related proteins), respectively [1,2,91,99,104,105]. They are important regulators of melanogenesis and melanocytic behavior of normal and malignant melanocytes [3,5,105,106,107,108,109,110,111,112,113,114]. Among the low penetrance genes are: *TYR* encoding tyrosinase (rate limiting enzyme of melanogenesis), *TRP1* encoding tyrosinase related protein 1 (melanogenic enzyme and protein stabilizing tyrosinase), *OCA2* encoding P protein (assists in the trafficking and processing of tyrosinase, transport of tyrosine transport, and regulation of melanosomal pH and glutathione metabolism), and agouti signaling protein (*ASIP*) acting as an endogenous antagonist of the melanocortin-1 receptor (MC1R) [1,4,100,112,113,115,116,117]. Mdm2, the product of the *Mdm2* gene is a negative regulator of the p53 tumor suppressor protein, while RB1 encodes retinoblastoma protein regulating G_1_ cell cycle arrest. In addition, several other low penetrance loci are connected to melanoma probability by genome wide association studies (GWAS) including *PLA2G6* (involved in regulation of pigmentation), *CASP8*, *AGR3*, *FTO* (affecting nevi formation), *PARP1*, *ATM* (associated with DNA repair), *CDKAL1*, and *CCND1* (methylthiolation of tRNA and regulation of cell cycle progression, respectively). Others include *ARNT/SETDB1*, *CYP1B1*, *MX2*, and *TMEM38B/RAD23B* (all discussed in [91]). Another locus for which mutations are associated with increased probability of melanoma is *VDR* [118,119,120,121,122,123,124] encoding the vitamin D receptor, a regulator of multiple phenotypic traits [69,70,125] that will be discussed in the following subchapters.

Malignant transformation of melanocytes and further progression to advanced stages of melanoma are driven by environmentally induced somatic mutations in a cellular, tissue, and systemic context-dependent fashion as discussed in numerous reviews [1,2,4,6,10,11,18,30,36,37,74,76,77,82,86,126,127,128,129,130]. Therefore, our overview on this topic will be brief. The most common mutations in CSD and non-CSD melanomas affect *BRAF*, *NRAS*, and *NF1* genes governing proliferation through action on the MAPK pathway, with BRAF and NRAS defined as oncogenes and NF1 as tumor suppressor gene. Of note, mutations in *BRAF* and *NRAS* are exclusive [131] and mutations in *BRAF* are very common in benign melanocytic nevi. Others include protein, phosphatase and tensin homolog (*PTEN*), and *KIT*, affecting cell metabolism, survival, and proliferation, *ARID1* and *2* (AT-rich interaction domain 1 and 2) affecting cell identity, *TP53* (tumor suppressor gene affecting resistance to apoptosis), *TERT* affecting replicative lifespan, and cell cycle control gene *CDKN2A*. There are several other mutations affecting melanoma progression in cutaneous melanoma exposed to sun [1,2,4,6,10,11,18,30,36,37,74,76,77,82,126,127] from which desmoplastic melanoma differ not in mutational load but in genes affected [36,132]. Acral melanomas, having no direct association with sun exposure, although having mutations in *BRAF*, *NRAS*, *NF1*, and *KIT*, show a lighter and different mutational burden in comparison to low-CSD or high-CSD melanomas [6,10,31,32,79,133,134,135,136]. In addition, recent transcriptomic analysis showed significant differences in gene expression and regulatory pathways between acral and non-acral melanomas in an Asian population [35].

### 2.2. Epidemiology and Risk Factors

Cutaneous melanomas predominantly affect people of European descent, with a significant risk being exposure to solar radiation, while melanomas of acral skin predominantly affect dark-skinned populations and are UVR independent [2,6,17,19,21,29,30,48,137,138,139]. It is expected that about 97,610 new melanomas will be diagnosed in the USA this year alone, including 58,120 in men and 39,490 in women, with 7990 people expected to die of melanoma, comprising 5420 men and 2570 women [140]. The lifetime risk of getting melanoma for White, Black, and Hispanic populations is, respectively, about 2.6%, 0.1%, and 0.6% in the USA [140]. Melanomas on skin intermittently exposed to sun frequently carry somatic *BRAF* mutations and have a higher nevus count, while melanomas chronically exposed to solar UVR have a low nevus count and *NRAS* and *KIT* mutations [6,10,14,30,48,141]. However, their mutational burden differs from desmoplastic melanomas [10,36]. Despite the association of *NRAS*-mutant tumors with chronic sun damaged skin, it is reported that UV signature lesions (C>T and CC>TT) are present in a similar proportion of *NRAS*- and *BRAF*-mutant melanomas (The Cancer Genome Atlas, 2015). Bowman et al., 2021, reported a differential sensitivity of *BRAF* and *NRAS*-mutant melanocytes to UV-mediated carcinogenesis in vivo. Their data suggest that a single UVB exposure triggers a greater burden of mutations in murine tumors driven by oncogenic BRAF than NRAS in pre-clinical murine melanoma models [142], Reduction of sun exposure and protection against solar radiation are recommended as preventive measures against melanomagenesis [6,30,40].

Other risk factors for melanoma development include xeroderma pigmentosum, the presence of atypical nevi, increased nevi count, a history of previous melanoma, previous non-melanoma skin cancer, redhead phenotype, immunosuppression, history of sunburn, male sex, a first degree relative with melanoma, blue eyes, age, use of indoor tanning beds, and finally the total exposure to sun listed as the lowest risk factor [30]. It must be noted that while low-CSD melanomas can arise in atypical dysplastic nevi, the majority of melanomas arise de novo, which is more visible in high-CSD melanomas [2,30]. As relates to skin type, the presence of pheomelanin represents a risk factor because of the mutagenic activity of its precursors and derivatives [4,8,84,143,144].

While it is unquestionable that exposure to sun increases the risk of melanoma, it must be noted that high-CSD melanomas, such as lentigo maligna type and desmoplastic melanoma, show relatively less aggressive behavior in comparison to low-CSD melanomas including superficial spreading type and nodular melanomas [145]. Berwick has even postulated that in patients having diagnosed melanoma, sun exposure was associated with increased patient survival [146], which has initiated a discussion on the link between UVB-induced production of vitamin D and possible benefit for melanoma patients [147,148].

Other factors involved in melanoma development are less defined and can include chronic inflammation [149,150,151], physical trauma [34,152,153], or viral etiology [154]. Although obesity (BMI ≥ 30) has been associated with increased melanoma incidence [149], a recent detailed epidemiological study in different states of the USA indicates lower melanoma incidence in obese Caucasians [155]. These authors speculate on a link between energy balance and antitumor immune responses or the likelihood that people with obesity spend less time outdoors walking or exercising and are more likely to wear covering clothing. The role of the microbiome in cancer including melanoma was recently emphasized [156] and detection of HPV in melanomas is intriguing [154,157] taking into consideration older work on transplantable melanomas [158].

### 2.3. Classification and Diagnosis

As mentioned in the introduction, cutaneous melanomas are subdivided into low- and high-CSD tumors and those unrelated to sun exposure, including acral, subungual, and mucosal (oral, genital, and anal) melanomas [2,21,30,135,152,159]. The classification is still based on histopathology and includes (1) superficial spreading type for low-CSD, (2) lentigo maligna and desmoplastic melanomas for high-CSD, (3) nodular and nevoid melanomas for various CSD melanomas, (4) melanomas with no sun exposure or without known etiological associations with sun exposure such as mucosal, acral, and Spitzoid melanomas, and (5) melanomas arising in blue or congenital nevi, pigmented epithelioid melanocytoma (PEM, animal type melanoma), and unclassifiable melanomas [2,21,30,152,160,161,162]. The diagnosis is made based on skin gross morphology, dermatoscopy, reflectance confocal microscopy (RCM), and finally evaluation of hematoxylin and eosin (H&E)-stained slides from the biopsy, with additional immunohistochemistry (IHC) if necessary. These represent the gold standards in the diagnosis of melanoma and atypical pigmented lesions. IHC includes evaluation for classical melanocytic markers such as SOX-10, MITF, S100, MART-1, and HMB45, together with markers of cell proliferation such as Ki-67 or PCNA, to mention the most important. In addition, preferentially expressed antigen in melanoma (PRAME) is used clinically to differentiate benign melanocytic nevi from melanoma [163,164].

Additional tools in the diagnosis and classification of primary melanomas are in situ fluorescence hybridization, comparative genomic hybridization (GH), gene expression profiling (GEP), and next-generation sequencing (NEGS). Also to be considered by the diagnostic dermatopathologist are transcriptomic, proteomic, and metabolomic approaches from viable fragments of biopsy or relatively non-invasive adhesive patch testing. Thus, the future of the diagnosis and classification of primary melanomas is in molecular markers with ancillary bioinformatics approaches, unless diagnosis is evident in H&E-stained sections. As detailed above, inherited and acquired genetic markers of melanoma should help in the classification and prediction of melanoma behavior [2,10,11,14,25,30,38,44,45,48,74,77,79,91,126,133,135,136,165,166,167,168]. These, together with multiomic analyses and bioinformatics and system biology approaches, should increase the precision of the diagnosis and prediction of the behavior of the lesion, and help identify non-genomic regulatory pathways and master regulators of melanomagenesis.

### 2.4. Staging and Current Therapy

#### 2.4.1. Staging

The staging of melanoma is based on decades of pathological analyses in relation to clinical outcomes, with the most important predicting factors being tumor thickness (Breslow depth) and anatomic invasion (Clark level) for melanomas localized to the skin [2,9,21,30,38,48,169,170,171,172,173]. Based on tumor thickness, the following tumor staging is applied (in parentheses tumor thickness in mm): pTO (no evidence of primary tumor), pTis (melanoma in situ), pT1a (<0.8), pT1b (0.8–1), pT2 (1–2), pT3 (2–4), and pT4 (>4), with an added a (no ulceration) or b (ulceration present). Other predictive factors that are part of synoptic reporting include the presence or absence of satellitosis, tumor-infiltrating lymphocytes (TIL), regression, angiolymphatic invasion, perineural invasion, radial vs. vertical growth phase, with pigmentation and cell morphology included by some pathologists. What is important to consider is the concept of radial (RGP) and vertical growth phase (VGP) with the latter indicating a gain in the metastatic capability of invading cells [9,171,172]. Transition from the RGP to the VGP with further metastatic phenotype is an example of tumor progression as defined by Foulds [174]. Of note, nodular melanoma represents a tumor at the pure vertical growth phase. Also, while tumor thickness represents a predictable marker of malignancy for the majority of cutaneous and mucosal melanomas, there are exceptions, including for desmoplastic melanoma, melanomas arising in blue or congenital nevi, or animal type melanomas. The above staging in the context of other predictive factors decides whether a sentinel lymph node biopsy (SLNB) is performed, usually for pT2 and higher. The pT stage affects clinical staging, with positive SLN overriding pT stage as a disease predictor. In general, clinical staging includes a tumor localized to the skin without histopathologic and clinical evidence of metastases: stages IA (pT1a), IB (pT1b or pT2a), IIA (pT2b, pT3a), IIB (pT3b, pT4a), and IIC pT4b). Stage III shows involvement of skin or lymph nodes (any pT tumor when melanoma spreads to local lymph nodes or there are satellites or in-transit melanoma) and stage IV (any pT tumor when melanoma metastasizes to distant lymph nodes and/or systemic organs). The pathological and clinical staging allows precise prediction of the natural history of the disease, including overall survival (OS) and disease-free survival (DF) times, and when supplemented by ancillary studies (IHC) or molecular testing (GP, GEP, NEGS) form a basis for an educated selection of appropriate therapeutic approaches. It must be noted that the staging of melanoma would likely be modified and/or supplemented by deposited RNA sequencing data, proteomic data, or metabolomic data at publicly available databases with implications for future therapy [1,11,18,30,48,74,75,76,77,79,127,130,134,159,165,169,175,176,177,178,179,180,181,182,183]. This will be further discussed in Section 5.5: Bioinformatics Considerations of Melanoma Diagnosis and Treatment.

#### 2.4.2. Therapy

The primary treatment of cutaneous melanoma includes surgical excision, which would be curable for in situ and thin melanomas, especially at the RGP. The OS decreases with increased tumor thickness, presence of ulceration, or other negative markers of melanoma behavior (see above). The margins of excision vary, depending on pT stage, anatomical site, and the country where the procedure is performed, and these recommendations have been evolving [30,38,48,170,184,185]. In the case of lentigo maligna melanoma, depending on the site of diagnosis and limitations for surgery, topical imiquimod (5%) [186,187] or radiation (in Europe) are offered [188]. In the case of a positive for melanoma sentinel lymph node biopsy (SLNB), some surgeons are offering elective lymph nodes dissection, which is, however, connected with increased morbidity and does not improve the OS [30,48,189,190]. For high risk of development of metastatic disease stages IIB and C, and when SLNB is positive, adjuvant postsurgical therapy can be applied with some limitations that either include immunotherapy using immune check point inhibitors (ICIs) or targeted therapy [30,48,191]. However, the majority of experts acknowledge that the serious adverse side effects may outweigh potential benefits for stage IIB/C and perhaps some IIIA patients (positive SLNB) vs. clinical observation [192]. This is in addition to the high costs of such therapies. The attendant toxicity of immunotherapy in melanoma patients is well documented [48,191,192,193]. Therefore, gene profile testing (GEP) promises to help which subsets of stage IIB/C would benefit from adjuvant therapy [192]. Interestingly, T-cell gene signatures derived from inflammatory skin diseases, such as systemic lupus erythematous (SLE) or psoriasis, can provide potential biomarkers for predicting the response to immune checkpoint blockade therapy in melanoma [194]. Stage III and IV melanoma patients are the subjects of immunotherapy, targeted therapy, or combined therapy [30,48,191,192,193,195,196,197].

Historically, conventional therapeutic approaches have included classical chemotherapy and radiotherapy, where applicable [41,42,48,198,199,200,201]. Chemotherapy includes drugs that inhibit melanoma cell proliferation or cause cell death. Resistance to these drugs is developed very fast and median survival time is very short. These drugs used to be the first line of treatment of advanced melanoma; however, now they are often replaced by immunotherapy and targeted drugs since they are more efficient and less toxic [202]. The most common drugs used that have been FDA approved were alkylating agents, including dacarbazine (DTIC), temozolomide, and fotemustine; taxanes, including nab-paclitaxel, paclitaxel, and docetaxel; and platinum agents, including cisplatin and carboplatin. Chemotherapy was used as monotherapy or in combination, such as paclitaxel and carboplatin [202,203,204] for advanced melanoma. In the past, radiotherapy was used in combination with chemotherapy after surgery, for stereotactic tumor destruction, or in combination with different therapeutic modalities [205,206,207,208,209].

The currently approved immunotherapy approaches involve targeting with antibodies the following ICIs: cytotoxic T lymphocyte-associated antigens 4 (CTLA-4), programmed cell death 1 (PD-1), PD ligand 1 (PDL-1), and lymphocyte-activation gene 3 (LAG3), with a combination of the antibodies if necessary, or in combination with other treatments [18,30,36,46,48,191,192,193,195,196,197,210]. They appear to be effective in a certain subset of patients [211]. Also, an adverse phenomenon of hyperprogression described in ICI therapy in solid tumors [212,213] is very rare in melanomas [214]. In the past, several immunotherapeutic approaches were tested, including the use of peptide from melanogenesis-related proteins, dendritic cells, tumor-infiltrating lymphocytes (TIL), adoptive T-cell therapy (ACT), interleukin 2, and interferons [215,216,217,218], none of which were sufficiently effective in phase 3 clinical trials [30,191]. Tumor-infiltrating lymphocyte (TIL) therapy for patients with advanced-stage melanoma have been reviewed in-depth recently [218]. It has to be mentioned that FDA recently approved TIL therapy (Iovance’s lifileucel from Amtagvi) for patients with unresectable or metastatic melanoma. The adoptive cell therapy in combination with other therapies, personalized melanoma vaccines, modified immune cells, or drugs targeting immune check points are currently being tested. Among them special attention is deserved by the effort to use oncolytic viruses in melanoma therapy [219,220] or mRNA-based melanoma vaccine [221,222,223,224]. In a recent trial, mRNA based on individualized neoantigen therapy (mRNA-4157 (V940)) was used in an open label, phase 2b study to treat high risk melanoma patients as an adjuvant with checkpoint inhibitor pembrolizumab compared with pembrolizumab alone [224]. Patients receiving combination therapy had longer recurrence-free survival and a lower rate of recurrence or death, together with a higher rate of metastasis-free survival after 18 months. There was no significant increase in toxicity with combination vs. monotherapy. Finally, it has been reported that vitamin D supplementation increased the objective response rate and prolonged progression-free time in melanoma patients undergoing anti-PD-1 therapy [59], opening up a window of opportunity to enhance existing therapy by a simple nutritional product supplement. This is further enhanced by a recent report that vitamin D supplementation may lead to fewer melanoma cases [60].

As relates to the targeted therapy use of BRAF inhibitors (if there is a *BRAF* mutation), this represents the standard care for metastatic melanoma both for melanomas associated with sun exposure or with lack of such association [18,30,37,48,136,176,191,196,225,226]. It must be noted that the most effective treatments are the use of BRAF inhibitors in combination with MEK inhibitors. BRAF as a target for melanoma and other solid tumors has recently been reviewed in detail [227]. The use of BRAF inhibitors in combination with ICIs, however, has shown unimpressive results when compared with targeted therapy alone [30,48,191]. Based on molecular analyses, alternative targeted therapies using NRAS and KIT inhibitors have shown some initial effect but are unimpressive as relates to the OS [30,48,191]. Similarly, an effort to target other pathways or using inhibitors of NF1, VEGFR, c-MET, and PI3K [30,48,86,175,177,178,191,195] remains to be proven for real effectiveness before being included into standard care therapy. It should be noted that targeted therapies are associated with high cost as well as side effects. Therefore, the challenge in current melanoma therapy is how to secure long-term survival in an economical manner with reduced side effects and improved patient comfort with therapy [228].

The developing resistance to targeted- or immuno-therapy leading to recurrent disease and death of the patient represents a significant challenge. There are different mechanisms underlying this negative phenomenon, including tumor heterogeneity with pre-existing mutations allowing clonal expansion of resistant cells, mutations during therapy and progression of the disease, mutator phenotype, metabolic heterogeneity, and dynamic changes in the tumor microenvironment, to name a few [4,6,25,30,37,74,75,87,130,165,178,179,182,211,218,229,230,231,232,233]. It must be noted that melanoma itself can affect the host response at local and systemic levels through production of neurohormonal regulators (as expected because of the neural crest origin of melanocytes) with immunosuppressive properties [3,50,82,234], including intermediates of melanogenesis and melanin that would increase resistance to any type of therapy [4,235,236]. Specifically, melanoma can produce various neuropeptides and neurohormones, such as POMC-derived MSH, ACTH and β-endorphin peptides, CRH, urocortins, enkefalis, TSH, TRH, neurotrophins, pro- and anti-inflammatory cytokines, catecholamines, serotonin, melatonin, and steroids including corticosterone and cortisol, as examples. These can affect not only melanoma behavior but also local and systemic host responses including global homeostasis depending which and in what quantity a given regulator is released into the circulation [3,50,82,88]. The tumor’s capability of regulating its own phenotype microenvironment and body homeostasis [82] makes the therapeutic battle for patient survival very challenging. The complexity of these processes requires combined, individualized, and sophisticated strategies with application of AI, to combat the development of tumor resistance to therapy.

## 3. Melanin, Melanogenesis, and Solar Radiation in a “Nutshell”

The main role of the melanocyte is to produce melanin pigment that protects against the harmful effects of solar radiation, although with some limitations, as described in many reviews and textbooks [3,115,237,238,239,240,241,242,243,244,245,246,247,248,249,250,251]. Melanin pigment and intermediates of melanogenesis can also play an important role in melanomagenesis, tumor progression, and responsiveness to therapy, as recently discussed [4]. Because of these considerations a short background on the principles of melanogenesis and its regulation is provided below.

Solar radiation represents a major cutaneous stressor as a stimulator of melanin pigmentation, and it acts as a full carcinogen in melanomagenesis, as discussed in several reviews [40,80,115,144,249,252,253,254,255,256,257,258,259,260]. However, the UVB spectrum also has a beneficial effect related to the production of vitamin D [69,72,261,262], plus there are homeostatic effects of UVR independent of vitamin D production [263]. Therefore, the subject deserves a short overview.

### 3.1. Solar Radiation: An Overview

UVR encompasses UVA (γ = 315–400 nm), UVB (γ = 280–315 nm with γ < 290 nm absorbed by the stratosphere), and UVC (γ = 200–280 nm, filtered by the ozone layer). Biologically relevant are UVB and UVA with the former being 1000 more effective than UVA in inducing phenotypic effects, being absorbed by the epidermis and the papillary dermis [246,264,265,266], while the later penetrates to the deep levels of the reticular dermis [246,264]. UVB and UVA exert different but partially overlapping mechanisms of action. UVB effects are predominantly secondary to absorption of its energy by chromophores, while UVA effects predominantly depend on oxidative changes induced by reactive oxygen species (ROS) [249,263,265,267,268]. Important UVB chromophores include pyrimidines and purines and their derivatives alone or in nucleic acids, trans-urocanic acid, quinones, indoles, compounds containing a benzene ring, melanins, unsaturated lipids, and 7-dehydrocholesterol (7DHC). Also relevant to melanomagenesis is that after absorption by the DNA, UVB induces covalent bond formation between adjacent pyrimidines leading to the production of mutagenic photoproducts such as cyclobutane pyrimidine dimers (CPD) and pyrimidine–pyrimidine adducts [253,256,269,270]. Oxidative damage mediated predominantly by UVA, and to some degree by UVB, includes oxidative damage of nucleotide bases such as oxidizing guanine to produce 8-hydroxy-2′-deoxyguanine (8-OHdG). These modifications lead to DNA mutations representing a major causative factor for melanoma [1,2,30,37,80,249,252,255,256,260,271]. Paradoxically, eumelanin, while protecting from UVR, can be involved in DNA damage after exposure to UVR through different mechanisms [144,250,257,272,273], which is aside of pheomelanin’s mutagenic activity [8,144,240,274], and it is necessary for UVA-induced melanomagenesis [254]. It should be noted that UVR, and UVB in particular, can induce local production of several molecules, including immunosuppressive cytokines, growth factors, neurohormones and neuropeptides, such as POMC-derived peptides, or corticosteroids that by affecting the local environment and activity of melanocytes would promote melanomagenesis and contribute to tumor progression [82,83,234,275,276,277,278].

However, UVB can also indirectly have anticancerogenic effects after absorption of its energy by 7DHC, which then undergoes transformation to vitamin D or lumisterol as described later in 4.1 [262]. Following activation, vitamin D and lumisterol compounds exert anti-inflammatory, anticancer, anti-oxidative, and DNA protective properties through diverse mechanisms [51,62,64,66,271,279,280,281,282,283,284,285]. While vitamin D signaling will be discussed in depth below, it is important to note that the most optimal UVB wavelength for pre-D3 production is at 295 nm, with negligible formation at >315 nm but still appreciable production at the longer UVC range [262,286].

### 3.2. Biochemistry of Melanogenesis

In humans, two types of melanin are produced: eumelanin with an indolic backbone and pheomelanin requiring only cysteine to produce benzothiazine and benzothiazole units [4,240,245,287]. The rate limiting reaction in melanogenesis is hydroxylation of L-tyrosine by the tyrosine hydroxylase activity of tyrosinase, with further L-DOPA oxidation (catecholase activity) to dopaquinone [237,248,288,289,290,291,292], and other reactions proceeding spontaneously with the rate of transformation defined by pH and the presence of cations [240,290,293,294,295] with possible further modification by additional enzymes [294,296,297,298,299,300,301]. It must be noted that phenylalanine hydroxylase can also affect melanogenesis by producing the L-tyrosine substrate for melanin pigment [5]. Tyrosinase also dehydrogenates 5,6-dihydroxyindole (DHI) [288] with gene mutation leading to oculocutaneous albinism 1 (OCA1). Non-enzymatic bioregulatory functions for tyrosinase or its alternatively spliced forms were also proposed [5,302]. Other characterized enzymes include tyrosinase related proteins type 1 and 2 (TRP1 and TRP2) [298,299,300] in which mutations lead to *OCA3* and *OCA8*, respectively [112]. Although mammalian TRP1 was recognized for its ability to oxidize DHI-2-carboxylic acid (DHICA) to hydroxyIndole-5,6-quinone-carboxylic acid, its function in the human system is less defined [112], except its role in stabilization of tyrosinase, regulation of its activity, and maintenance of the proper melanosomal structure ([112,115,303,304]). TRP1 also can regulate melanocyte proliferation and prevent cell death ([112,115,303]) with non-coding *TYRP1* mRNA promoting melanoma growth [305]. TRP2 shows DOPA tautomerase activity transforming dopachrome to DHICA [298,300,306].

It should be noted that in vivo, melanogenesis takes place within membrane bound organelles, melanosomes, because of the high toxicity of the intermediates of melanogenesis. Thus, the final steps of production of melanin pigment are preceded by synthesis not only of tyrosinase and related proteins but also of other melanosomal proteins and melanogenesis-related proteins with their processing by rough endoplasmic reticulum, transport to melanosomes, activation of melanogenesis, and regulation of its velocity in these organelles [115,290,307,308,309,310,311,312]. This includes the acquiring of copper by tyrosinase and its transport to the melanosome, copper being a metal cofactor required for its enzymatic activity [313,314], while zinc is required for TRP1 [315] and TRP2 [314]. To complete the description of melanogenesis it is also important to mention alternatives to tyrosine and DOPA substrates for melanin synthesis, such as tryptophan [240,316] or oxidation of catecholamines to produce neuromelanin [317], which can have a protective or cytotoxic role depending on the context [318,319]. Finally, in the skin, melanocytes transfer melanosomes via dendrites to keratinocytes [309,310,320,321]. In keratinocytes the melanosomes form a visible “cap” above the keratinocyte nucleus, providing some protection from UV damage to these cells [244,246,247,310,320,322]. However, other investigators indicated that melanosomes can act as metabolically active organelles with the potential to regulate keratinocytes and epidermal functions through different mechanisms [4,247,323,324,325].

### 3.3. Molecular and Hormonal Mechanisms Regulating Melanin Pigmentation

Other structural melanosomal proteins involved in the intracellular regulation of melanin synthesis include SILV/PMEL17/GP100 and MLANA/MART1 [112,310,326], which also serve as histochemical markers of melanoma [326]. Intracellular regulatory proteins include transporter proteins, such as solute carrier family 45 member 2 (SLC45A2) and SLC24A5, which is also involved in regulation of intramelanosomal pH in which mutations lead to OCA4 and OCA6, respectively [112,114], oculocutaneous albinism 2 protein/P protein (OCA2) that assists in the trafficking and processing of TYR, tyrosine transport, the regulation of melanosomal pH, glutathione metabolism, and anion transport [112]. Others include leucine-rich melanocyte differentiation associated (LRMDA) (OCA7), G protein-coupled receptor 143 (GP143), and RAB27A [111,112,327].

MITF is a master regulator of melanogenesis-related genes, as well as controlling other genes involved in the regulation of differentiation, proliferation, and apoptosis [1,3,105,278]. Its expression and functions are regulated by SRY-related HMGbox (SOX) 9, SOX10, cAMP response element-binding protein (CREB), PAX3, lymphoid enhancer-binding factor 1 (LEF1), zinc finger E-box binding protein 2 (ZEB2), one cut domain 2 (ONECUT2), and MITF itself. The list of cell surface receptors involved in regulation of melanocyte activity and melanin pigmentation includes G-protein coupled receptors (GPRS) such as melanocortin receptor type 1 (MC-1) which is a target for POMC-derived MSH peptides (α, β and γ), and ACTH or its fragments [1,3,109] after being processed by convertases PC1 and PC2 in the skin cells including melanocytes [234,328]. The MC-1 antagonists are agouti signaling protein (ASIP) and β-defensin 3 (HBD3), which by inhibiting the action of MC1 agonists decrease melanogenesis with a switch toward production of pheomelanin [112,329,330,331]. Another GPCR, endothelin receptor type B (EDNRB), a target for endothelins, plays an important role in melanocyte development and differentiation, but has also been associated with melanoma progression [332,333,334]. Other GPCRs involved in the regulation of melanogenesis and melanocyte functions are: µ-opiate, a target for β-endorphin [335,336], corticotropin releasing hormone receptors (CRHRs) [234,337], β2-adrenoreceptors (β-AR) and α1-AR, targets for catecholamines [338,339], and glutamate receptor 1 (mGlUR1) regulating melanocyte proliferation and stimulating melanoma cells [195,275]. Other important regulators include transient receptor potential cation channel, subfamily M, member 1 (TRPM-1) which also inhibits melanoma progression [340,341], and KIT protein representing tyrosine kinase receptor important for melanocyte development and distribution in the skin, which plays a role in melanoma development [332,342]. There are also other receptors activated by different signaling molecules, including histamine, serotonin, melatonin, acetylcholine, steroids, and cytokines, which can be involved in the positive or negative regulation of melanin pigmentation and are reviewed in [3,82,83,98,112,275,316,327,343,344,345,346,347].

All of these regulators with their cognate receptors can regulate not only melanin pigmentation but also the activity of normal and malignant melanocytes and affect melanoma behavior, as discussed recently [82]. Educated targeting of some of these regulatory networks or understanding their impacts can be utilized when choosing the optimal therapy for melanoma and cancer in general [50,82,348].

The POMC signaling system deserves special attention because of its role in the regulation of melanin pigmentation [3,115] and its widely recognized role in the body’s responses to stress [349,350,351,352]. It is also a part of the endogenous skin responses to environmental stress securing the homeostatic integrity of this organ [83,328,353]. However, because of the pleiotropic properties of POMC-derived peptides, including their direct or indirect immunosuppressive effects, when this system is deregulated during melanoma progression it can negatively affect the disease outcome [50,82]. In fact, increased expression of POMC, its peptides, and its regulators were noted during melanoma progression [234,354,355,356]. Thus, the local POMC signaling system with its upstream and downstream regulatory pathways [83,234,328,348,349,357,358] can represent a target either for melanoma prevention or adjuvant therapy in a context dependent fashion [82].

### 3.4. Complex Role of Melanin Pigment and Melanogenesis in Melanoma Progression and Therapy

As mentioned above, eumelanin protects against UVR-induced melanoma development [115,247], while also contributing to melanoma initiation [254,359]. On the other hand, pheomelanin is a risk factor for melanoma development because of its instability and mutagenic activity with or without exposure to UVR [4,8,84,143,144,245,273,274]. It should also be noted that pigmentary activity is also regulated by the POMC signaling system [3,109,115] with a feedback mechanism envisioned [3,4].

Most recently, the Yin and Yang role of melanin in melanomagenesis, melanoma progression, and therapy were discussed in detail (See Slominski et al. 2022) [4] (Figure 1); therefore, we will focus briefly on the most clinically relevant aspects of melanogenesis affecting melanoma disease outcome. Clinicopathological analyses revealed that while at stage I melanoma, positive regulation of melanin synthesis may have a beneficial effect for melanoma patients [340,341]. In advanced melanomas, melanogenesis leads to shorter overall survival (OS) and disease-free survival (DSF) times [360] and the presence of melanin negatively affects the therapeutic outcome [361]. This is further exemplified by the negative effects of melanin pigmentation on disease outcome in uveal melanomas [7,362,363]. However, we note histopathological studies on a German population where pigmentation did not appear to provide significant additional value for prognostic assessment pT classification [364]. In contrast, recent omics analysis of the available database by Bakr et al., 2022, showed poorer prognosis for melanogenesis signature [183]. The increased resistance to therapy of pigmented melanoma cells is well known and supports the above clinicopathological observations [235,238,365,366,367,368,369,370,371,372,373,374,375,376]. In addition, products and intermediates of melanogenesis are not only cytotoxic [248,295,377,378,379,380] but also highly mutagenic and immunosuppressive [235,295,371,373,381,382,383,384]. Furthermore, melanin pigment and the process of melanogenesis consume oxygen [115,237,243,247,288,385,386,387,388], potentially generating a hypoxic environment within the cell. In fact, stimulation of melanogenesis can affect intracellular metabolism [389,390,391,392] and stimulates HIF-1α signaling [236]. Thus, melanogenesis through diverse mechanisms can affect skin functions [3,5,324] and melanoma development and behavior (Figure 1) [4]. Thus, inhibition of melanogenesis represents a realistic adjuvant strategy to amplify existing therapeutic approaches (Figure 1) [4,235,367,393]. In this context, it is worth mentioning that a diverse range of non-toxic inhibitors of tyrosinase activity are available [292,394,395,396], which in addition to copper chelators (cofactor for tyrosinase) would inhibit tyrosinase and decrease melanin synthesis [365,367,371,373,397,398,399,400]. However, we acknowledge that others, including some of us, have proposed using the melanogenic pathway as a tool for melanoma therapy utilizing the cytotoxicity of melanogenesis products [248,377,380,401,402], also reviewed in [4]. However, this strategy has not been validated in the clinic. In contrast, older data have shown the opposite results, including a positive effect of diet restriction on melanogenesis precursors on melanoma outcome in human and animal models [403,404,405].

Lastly, immunocytochemistry studies on human melanoma samples have shown that high levels of melanin pigmentation are associated with decreased expression of the vitamin D receptor (VDR) [406], the CYP27B1 enzyme that is involved in vitamin D activation [407], and retinoic acid-related orphan receptors (ROR)α and γ [408] on which vitamin D hydroxyderivatives can act as inverse agonists [409]. However, cell culture studies on rodent melanomas have shown a complex effect of melanin pigmentation on vitamin D signaling, with cells showing moderate levels of pigmentation being sensitized to the antiproliferative effects of vitamin D hydroxyderivatives in comparison to the amelanotic phenotype, but with heavy melanization levels leading to reduced expression of the *VDR*, *RXR*, *CYP24A1*, and *PDIA3* mRNA and a decrease of the ligand-induced VDR translocation to the nucleus [410]. In vitro studies on human melanomas have shown that pigmented phenotype made melanoma cells resistant to anti-melanoma activity of vitamin D hydroxyderivatives, which was dependent on their inhibition of the nuclear factor kappa-light-chain-enhancer of activated B cells (NF-κB) activity [411]. Thus, the melanogenesis/melanin-induced mechanism underlying vitamin D signaling in melanoma cells is complex and context dependent. Of key interest would be the clinicopathological findings on interactions between HIF-1α (its signaling is upregulated by melanogenesis [236]) and VDR or ROR α and γ [57], and the relationship between expression of NF-κB (target for vitamin D active forms) and melanogenic activity [411]. Defining these complex interactions would require in-depth analyses of deposited omics data with the help of the latest bioinformatic tools in relation to melanoma behavior, disease staging, and finally the clinical outcome.

## 4. Cutaneous Vitamin D in a “Nutshell”

### 4.1. Production, Activation, and Receptor Targets

#### 4.1.1. Activation of Vitamin D

Vitamin D3 is photoproduced in the skin from the action of UVB radiation on 7-dehydrocholesterol (7DHC) [261,286,412,413,414,415,416] which is a final intermediate in the cholesterol biosynthetic pathway [73,417,418,419]. The UVB breaks the bond between carbons 9 and 10 of the B-ring of the 7DHC-producing previtamin D3, termed a secosteroid due to the ring cleavage [71,286,412]. Once formed, previtamin D3 undergoes thermal isomerization at skin temperature to produce vitamin D3 [71,286,412]. Vitamin D3 can be released from the skin and carried in the bloodstream bound to vitamin D binding protein (DBP) or albumin [412,420]. In the liver, vitamin D3 undergoes the initial step in its activation, hydroxylation at C25. This is primarily carried out by CYP2R1, a microsomal cytochrome P450 enzyme, and produces 25-hydroxyvitamin D3 (25(OH)D3) [421,422,423]. This C25 hydroxylation can also be catalyzed by CYP27A1, a mitochondrial cytochrome P450 involved in bile acid synthesis [422,423,424]. The 25-hydroxyvitamin D3 is released from the liver where it is the major circulating form of vitamin D3 and is carried in the bloodstream bound to DBP [420]. It undergoes the final step in its activation, 1α-hydroxylation by CYP27B1, in the kidney, producing the hormonally active form of vitamin D3, 1,25-dihydroxyvitamin D3 (1,25(OH)_2_D3) [420,425,426]. The kidney is also the major site of inactivation of both 25(OH)D3 and 1,25(OH)D3 catalyzed by CYP24A1 [420,425]. This enzyme is induced at the level of transcription by high concentrations of 1,25(OH)_2_D3. The major initial step in the inactivation is hydroxylation at C24 producing 24-hydroxyvitamin D3 or 1,24,25-trihydroxyvitamin D3, the former being found at appreciable levels in the plasma and serving as a marker of CYP24A1 activity [66,420,427]. The initial hydroxylation at C24 is followed by a series of CYP24A1-mediated oxidations of the vitamin D3 side chain, ultimately producing the water-soluble C23 carboxylic acids which are excreted [66,420,427]. The above classical pathways of vitamin D3 activation and inactivation are not completely confined to the liver and kidneys, occurring in a range of other tissues including the skin [412,428,429,430,431,432,433,434,435], leading to autocrine and paracrine actions of 1,25(OH)_2_D3 [421,425,426,436,437]. Normal melanocytes and melanoma cells express CYP2R1, CYP27A1, CYP27B1, and CYP24A1 at variable levels, and so can potentially activate and inactivate vitamin D3 by the above pathways [407,438,439,440,441,442,443], which in fact was demonstrated in cultured melanoma cells [444].

It is now well established that there is an alternative pathway of vitamin D3 activation initiated by the action of CYP11A1 [345,445], best known for catalyzing the first step in steroid hormone synthesis [446,447], the cleavage of the side chain of cholesterol-producing pregnenolone (Figure 2). As shown in Figure 2, CYP11A1 hydroxylates the side chain of vitamin D3, primarily producing 20-hydroxyvitamin D3 (20(OH)D3), 20,23-dihydroxyvitamin D3 (20,23(OH)_2_D3), and 20,22-dihydroxyvitamin D3 (20,22(OH)_2_D3), plus other minor products, but does not cleave the vitamin D3 side chain (reviewed in [73,420,448,449]). These initial products can undergo 1α-hydroxylation catalyzed by CYPB1, producing 1,20(OH)_2_D3 in the case of 20(OH)D3 [450,451,452]. These pathways were initially discovered in studies with purified recombinant enzymes [453,454,455,456] but were confirmed using isolated adrenal and placental mitochondria and cultured cells, including keratinocytes and dermal fibroblasts [435,451,455]. More recently, both 20(OH)D3 and 1,20(OH)_2_D3 have been measured in the plasma of 103 healthy participants at mean concentrations higher than that for 1,25(OH)_2_D3 [457], confirming that these pathways occur in vivo. The CYP11A1-derived secosteroids display biological activity (see later) but have no (20(OH)D3 and 20,23(OH)_2_D3) or limited (1,20(OH)_2_D3) calcemic activity compared to 1,25(OH)_2_D3 [458,459,460]. The major CYP11A1-derived hydroxyderivatives can also be metabolized by CYP24A1, which catalyzes their hydroxylation at C24 or C25, and can increase their potency rather than decrease it, at least for inhibition of melanoma cell proliferation [420,461]. CYP11A1 is also expressed in normal melanocytes and melanoma cells [441,454], so these cells can potentially carry out the CYP11A1-initiated pathways of vitamin D metabolism, with subsequent modification of products by CYP27B1 [407,439], CYP24A1 [438,439], or CYP27A1 [439,440], which are also expressed in melanoma cells [439,441].

#### 4.1.2. Activation of Lumisterol and Tachysterol

Lumisterol3 (L3) and tachysterol3 (T3) are derived from the action of UVB radiation on previtamin D3 and can also be activated by CYP-dependent pathways (Figure 2). These sterols are considered as over-irradiation products and arise from the absorption of UVB energy by previtamin D3 during prolonged sunlight exposure, which causes B-ring resealing to produce L3, or isomerization to produce T3 [71,286,412]. L3 is a stereoisomer of 7DHC, but unlike 7DHC, it cannot be acted on by 7-dehydrocholesterol reductase enabling retention of its 5,7-diene structure during subsequent metabolism [462]. L3 can be hydroxylated by CYP11A1 to produce 24(OH)L3, 22(OH)L3) and 20,22(OH)_2_L3 as the major products [463,464] whereas CYP27A1 acts on L3 to produce 25(OH)L3, (25*R*)-27(OH)L3 and (25*S*)-27(OH)L3 [465]. All of these products display biological activity on skin cells [64,66,464,465,466,467] (see later). Both CYP11A1 and CYP27A1 can also act on T3, the former producing 20(OH)T3 and the latter producing 25(OH)T3, with both products being active on skin cells [468]. Cleavage of side chain of L3 by CYP11A1 can produce pL, a potential substrate for steroidogenic enzymes [73,345]. Since CYP11A1 [454] and CYP27A1 [439,440] are expressed in melanoma cells, the above metabolites could be produced in these cells from locally available L3 or T3.

#### 4.1.3. Photoactivation of Products of 7DHC Side Chain Cleavage and Further Metabolism

CYP11A1 can metabolize 7DHC to 7DHP with 22(OH)7DHC and 20,22(OH)_2_7DHC serving as intermediates of the pathway [453,454] (Figure 2). These pathways operate in vivo [469,470], and their intermediates are detectable in human serum and tissues [471]. 7DHP can be metabolized by steroidogenic enzymes to Δ7-steroidal derivatives [469,470]. The absorption of UVB by the B ring of these Δ7 steroids leads to their transformation to the secosteroidal configuration (Figure 2), as originally proposed [449,454] and experimentally confirmed [472,473,474]. The majority of these compounds are biologically active [449,458,469,470,473,474,475,476,477,478]. Their effects on melanoma are discussed below.

#### 4.1.4. Receptor Targets for Active Forms of Vitamin D, Lumisterol and Tachysterol

The proposed targets for D3 and L3 after metabolic activation are shown in Figure 3 with potential phenotypic outcomes. The role of the vitamin D receptor (VDR) in mediating the actions of 1,25(OH)_2_D3 is well established [68,125,479,480]. The hydroxyvitamin D3 derivatives produced by CYP11A1 can also bind to the VDR [481,482,483,484], acting as biased agonists, displaying some but not all of the biological effects of 1,25(OH)_2_D3 [485,486]. Unlike 1,25(OH)_2_D3, metabolites lacking a 1α-hydroxyl group, such as 20(OH)D3, 20(OH)D2, and 20,23(OH)_2_D3, have little or no calcemic activity [458,459,487,488], but exhibit many of the other VDR-mediated effects seen for 1,25(OH)_2_D3, including anticancer effects [487,489,490]. Analysis of gene expression profiles has demonstrated that 20,23(OH)_2_D3 can also act via the aryl hydrocarbon receptor (AhR), supported by molecular modeling studies which included its downstream metabolites [483,486,491]. 20(OH)D3, 1,20(OH)_2_D3, and 1,25(OH)_2_D3 displayed lesser abilities to act via the AhR compared to 20,23(OH)_2_D3 [483,486,491].

Both 20(OH)D3 and 20,23(OH)_2_D3 and their downstream metabolites have been identified as inverse agonists of RORα and RORγ [409,483,485], reducing basal receptor activity and thus displaying anticancer, anti-inflammatory, and antifibrotic activities [409,485,492,493,494]. These receptors are expressed in skin, including melanoma cells [408,409]. Recently, it was reported that the CYP11A1-derived hydroxyvitamin D3 derivatives also act via liver X receptors α and β (LXRα and LXRβ), which are expressed in the skin and for which a number of naturally occurring oxysterols act as ligands. 20(OH)D3 and 20,23(OH)_2_D3 were observed to act as agonists on both forms of the receptor, whereas derivatives with a 1α(OH) group such as 1,20(OH)_2_D3 and 1,25(OH)_2_D3 act as agonists on LXRβ, but as inverse agonists on LXRα [483,495]. Peroxisome proliferator-activated receptor γ (PPARγ) is also considered as the receptor for CYP11A1-derived hydroxyderivatives of vitamin D and of hydroxyderivatives of lumisterol [496] (paper in preparation) and tachysterol [468]. It should be noted that VDR-independent mechanisms of action have also been indicated for 1,25(OH)_2_D3 [497,498,499,500,501,502,503]. In addition, receptor independent mechanisms of action for vitamin D3 hydroxyderivatives were recently suggested [504,505].

The CYP11A1- and CYP27A1-derived hydroxylumisterols can also act via the VDR, but molecular modeling and functional studies indicate that this is via the nongenomic site of the VDR (A-VDR) [464], previously identified as the site of binding of the synthetic lumisterol derivative, 1,25-dihydroxyL3 (1,25(OH)_2_L3), that mediates rapid effects [506,507]. The reduction of UV-induced damage to skin by 24(OH)L3, shown to be mediated by the VDR by knockdown experiments, appears to involve this non-genomic pathway [490]. There is also evidence for involvement of endoplasmic reticulum stress protein, ERp57, also known as PDIA3, in this process [490]. Molecular modeling and functional assays indicate that the hydroxylumisterols can act as inverse agonists on RORα and RORγ, similar to CYP11A1-derived hydroxyderivatives [464]. The CYP11A1- and CYP27A1-derived hydroxylumisterols also act as agonists on LXRα and β [495] and on AhR and PPARγ [496]. Lastly, the tachysterol derivatives 20(OH)T3 and 25(OH)T3 have been reported to act as agonists on the VDR, AhR, LXRα and -β, and PPARγ, with experimental data supported by molecular docking studies. These studies, plus robust stimulation of CYP24A1 expression, suggest that the hydroxyT3 derivatives act via the genomic site of the VDR rather than the non-genomic site, in contrast to the lumisterol derivatives [468].

### 4.2. Action of Vitamin D Active Forms on Normal Melanocytes

#### 4.2.1. Proliferation and Melanogenesis

The regulation of melanogenesis by active forms of vitamin D has been controversial for decades [3]. Thus, stimulation of melanin pigmentation in B16 melanoma [508] and of tyrosinase activity in human melanoma cells [509] and normal melanocytes [510] was reported. Abdel-Malek et al., 1988, reported that vitamin D precursor could enhance melanin pigmentation of the pinnal epidermis of DBA/2J mice in vivo with an increased number of DOPA positive melanocytes, but had no effect on normal human melanocytes, while 1,25(OH)2D3, and to a lesser degree 25(OH)D3, inhibited melanogenesis but had no effect on cell proliferation [511]. Other studies have shown a lack of effect of either provitamin D3, lumisterol, previtamin D3, vitamin D3, 25(OH)D3, or 1,25(OH)_2_D3 on melanogenesis in human melanocytes and S91 melanoma cells [512]. An inhibitory effect of 1,25(OH)_2_D3 on proliferation of normal human melanocytes was also reported [513]. We also tested the effect of 1,25(OH)_2_D3 and CYP11A1-derived 20(OH)D3, 20,23(OH)_2_D3, and 1,20(OH)_2_D3 on normal and malignant melanocytes [441] and found a lack of effect on melanogenesis. However, D3-hydroxyderivatives had strong inhibitory effect on melanoma proliferation with weaker effect on proliferation of normal melanocytes. In normal melanocytes, compounds with a hydroxyl group at Cα1 had stronger anti-proliferative effects than 20(OH)D3 and 20,23(OH)_2_D3, and inhibited dendrite formation, and the latter had no effect on this. Thus, future careful studies on melanocytes in cell culture from different racial background, in vivo studies in animal models, and ex-vivo studies on corporal or scalp skin, or 3D models, are necessary to establish a role for different vitamin D hydroxyderivatives in the regulation of melanogenesis and melanocyte behavior.

On the other hand, the role of constitutive or facultative melanin pigmentation appears to be well established, with the consensus that it inhibits cutaneous vitamin D production acting as a natural sunscreen, predisposing to many systemic pathologies [69,71,262,514,515,516]. However, this correlation is not linear [514,517] and aside from the sunscreen role would involve complex interactions, including interactions with genes and proteins regulating melanogenesis [518,519,520]. These inconsistencies can be explained by the optical properties of melanin where eumelanins in pigmented epidermis have an estimated sun protection factor (SPF) value of between 2–3 compared to depigmented epidermis, while pheomelanin has an even lower SPF [325,521]. Furthermore, melanin pigment has several bioregulatory functions in the skin [4,242,243,272,325,384,522,523] and precursors and intermediates of melanogenesis can act as hormone-like regulators [3,5,323] all defining melanocytes as sensory and regulatory cells of the epidermis, as originally proposed [324]. Thus, concerted actions of melanins, intermediates, and precursors of melanogenesis and their regulators could affect epidermal vitamin D3 photosynthesis and activation with melanocytes playing a central regulatory role. This represents an important subject worthy of future in-depth investigations.

#### 4.2.2. Photoprotective Effects on Melanocytes

The active vitamin D metabolite, 1,25(OH)_2_D3, has been shown to have photoprotective effects on human melanocytes and other skin cells [64,271,524,525,526]. Indeed, 1,25(OH)_2_D inhibited UVR-induced melanocyte cell death due to reduced UVR-induced DNA damage in the form of thymine dimers [524,525]. It also inhibited UVB-induced apoptosis in normal melanocytes [513]. Similar protective effects against UVB-induced damage on keratinocytes have been shown for CYP11A1-derived vitamin D3 and lumisterol hydroxy metabolites, which were associated with stimulation of p53 and NFR2 signaling pathways and downregulation of IL-17 and NFκB signaling with anti-inflammatory effects [64,65,73,466,527]. Moreover, in vivo studies have demonstrated that 1,25(OH)_2_D, applied topically, protects against DNA damage and sunburn cells in mice [507,525,528,529] in acute UVR models, and also inhibits chronic UVR-induced skin carcinogenesis in a mouse model [529]. Vitamin D analogs have also shown promise as photoprotective agents. A low calcemic synthetic lumisterol analog 1,25(OH)_2_lumisterol3 (JN) was protective against UVR-induced cell death and thymine dimers in melanocytes [525]. While 1,25(OH)_2_D is conformationally flexible, JN is a 6-s-*cis* conformationally restricted analog that can generate only non-genomic biological responses [529]. Another low calcemic analog, 1α-hydroxymethyl-16-ene-24,24-difluoro-25-hydroxy-26,27-bis-homovitamin D3 (QW), which has some transactivating capacity, has shown significant protection against UVR-induced thymine dimers in melanocytes and in hairless mice [525,529]. While JN significantly reduced UVR-induced squamous cell carcinoma-type tumor incidence and multiplicity in a chronic UVR model [525], QW was only effective in reducing tumor multiplicity [530]. QW has, however, been shown to protect against chemical-induced skin carcinogenesis and tumor latency [531]. In addition, photoprotective effects of 20(OH)D3 were shown in mouse skin and of 24(OH)L3 in human skin ex vivo with similar potency to 1,25(OH)_2_D3 [232,532]. Furthermore, experiments on mice in which melanocyte specific ablation of VDR was performed have shown increased damage in melanocytes of murine neonatal skin after exposure to UVB [281].

Exposure of skin cells including melanocytes to UVR results in increases in levels of tumor suppressor protein p53. Following exposure to UVR, increases in p53 can activate pro-apoptotic mechanisms to remove damaged cells, or can disrupt the cell cycle to allow time for DNA repair prior to replication. The protective effects of vitamin D compounds against UVR-induced damage in melanocytes were accompanied by further increases in levels of p53, which may in part account for the 1,25(OH)_2_D-induced reductions in UVR-induced DNA damage [525]. Moreover, studies by Shariev et al. demonstrated that UVR decreased levels of another tumor suppressor, PTEN, in primary human melanocytes and in Skh:hr1 hairless mouse skin [533,534]. Treatment of melanocytes with 1,25(OH)_2_D inhibited this UVR-induced depletion of PTEN [533], likely due to the 1,25(OH)_2_D-induced inhibition of AKT phosphorylation [279]. Loss of PTEN can result in suppression of a key nucleotide excision repair protein, xeroderma pigmentosum complementation group C protein (XPC). It has been demonstrated that 1,25(OH)_2_D can increase levels of XPC in human skin [467] and also that the VDR can bind the PTEN promoter in gastric cancer cells to inhibit apoptosis. PTEN has also been associated with melanoma development [534].

## 5. Role of Vitamin D in Melanoma

### 5.1. Serum Vitamin D Hydroxy Metabolites Levels and Melanoma Progression

An inverse relationship between vitamin D serum levels and cancer incidence has been recognized for decades, with a consensus that vitamin D deficiency can increase cancer incidence and cancer related mortality [71,535,536,537,538,539,540,541,542,543,544,545,546,547,548,549,550,551,552]. Similarly, many studies have pointed to an inverse relationship between melanoma progression and outcome, and to a certain degree the incidence, to serum levels of 25(OH)D used as a marker of vitamin D status [553,554,555,556,557,558,559,560,561]. Although a large meta-analysis found no relationship overall between vitamin D intake or serum 25(OH)D concentrations and risk of melanoma [562], there is increasing, reasonably consistent evidence from several countries that low concentrations of 25(OH)D (generally <50 nmol/L) around the time of melanoma diagnosis are associated with poor prognostic indicators, including higher Breslow thickness (tumor thickness) [553,556,557,559,561,562], increased ulceration, and higher mitotic rate [563]. Not surprisingly, clinical studies have consistently reported worse outcomes, such as increased progression, decreased recurrence rates, and increased death rates, over the following 3–5 years [553,564]. Timerman et al. [560] reported that patients with metastatic melanoma, who were vitamin D deficient (<50 nmol/L) on testing within one year of initial diagnosis and who subsequently had a decrease in their 25(OH)D serum concentration, or whose vitamin D status did not improve by more than 50 nmol/L, were more than 4.5 times more likely to have worse outcomes compared to patients with non-low 25(OH)D at baseline and who had a subsequent increase by more than 50 nmol/L. Other studies also indicated a casual association between serum levels of 25(OH)D3 and melanoma incidence [61]. A recent systematic review and meta-analysis has shown an association of increased incidence of melanoma with vitamin D insufficiency and a less favorable tumor thickness [565], supporting again a concept of inverse correlation between vitamin D levels and melanoma. Finally, recent studies have indicated that regular use of vitamin D is associated with fewer melanoma cases [60] and its supplementation increased objective response rate and prolonged progression-free time in patients with advanced melanoma undergoing anti-PD-1 therapy [59]. While there is consensus that vitamin D deficiency can negatively impact melanoma disease progression and outcome, the conflicting analyses on the levels of 25(OH)D and melanoma incidence require explanation. Specifically, serum levels of 25(OH)D3 mostly reflect the exposure of the skin to UVB, a factor contributing to melanomagenesis. Therefore, this confounding factor will affect any analysis on the relationship between cutaneous melanoma incidence and 25(OH)D levels, unless such studies are restricted to cases that are not related etiologically to sun exposure, such as acral or mucosal melanomas. How complex such relationship can be is illustrated by studies by Berwick et al. [146] showing that sun exposure is associated with increased survival of patients already having melanoma.

### 5.2. Role of Classical and Non-Classical Receptor Targets and Modifying Enzymes in Melanoma Progression

#### 5.2.1. Clinicopathological Studies

The phenotypic response of target cells and tissues to the anticancer action of vitamin D and its derivatives requires the presence of classical and non-classical receptors, VDR, RORα, and RORγ (Figure 4). Knocking down VDR expression resulted in the enhanced proliferation of melanoma cells grown in 2D and 3D models and higher migration potential [489]. In 2011, we published the first paper describing the changes in vitamin D receptor expression pattern in clinical samples (25 nevi, 69 primary cutaneous melanomas, 35 metastases, and five local recurrences) [406]. VDR was localized in the nucleus and cytoplasm of the cells, and VDR expression was higher in benign lesions and decreased in tumors, both primary and malignant. Moreover, in lesions showing the most malignant phenotype, vertical growth phases, nodular histotype, and in primary melanomas with highest Clark level (V) and Breslow depth (≥3.1 mm) there was significant decrease or loss of VDR expression. Cytoplasmic immunolocalization of VDR was also lower in melanomas that developed metastases than in non-metastasizing melanomas. The lack of VDR in primary melanomas was related to worse overall survival of melanoma patients. It should be noted that VDR expression was affected by melanoma pigmentation and the lower VDR level was accompanied by melanin presence [406]. Re-analysis our data and the inclusion of new data confirm the potential of VDR as a prognostic and predictive marker [58]. We observed a correlation between decreasing VDR expression and melanoma progression assessed according to pT stage. For overall stage 2–4, VDR expression was significantly lower than in stage 1. In addition, pM1 melanomas were accompanied by the lower VDR in comparison to pM0. Correspondingly, the presence of other pathological prognostic markers of unfavorable prognosis such as ulceration or absent or non-brisk tumor-infiltrating lymphocytes (TILs) was characterized by significantly lower VDR expression. The presence of VDR expression and the lack of ulceration delineated better overall survival than melanomas without VDR expression and with or without ulceration [58]. Some of these findings were confirmed by La Marra and colleagues [566] in an Italian cohort. Although the authors reported predominantly cytoplasmic staining of melanoma cells (caused by technical aspects of immunostaining, using different antibodies and characteristic of melanoma patients included to both studies), a lack of differences between metastasizing and non-metastasizing melanomas, and the presence of markers of worse prognosis (absence of TILs, presence of ulceration, mitotic index, VGP vs. RGP, and satellitosis), they observed higher VDR expression in stage I melanomas, superficial spreading histological type, and correlation with Breslow thickness and Clark levels [566]. A decreased expression of VDR in melanomas in comparison to benign melanocytic nevi was also reported by another group [567]. Immunocytochemical studies by Hutchinson et al. have shown that decreased nuclear VDR expression correlated with increased cytoplasmic staining, suggesting the failure of nuclear entry as a primary cause of defective VDR signaling in melanoma. Other studies have shown that higher expression of VDR in melanoma represents a good prognostic factor [54]. The defective VDR signaling in melanoma and its correlation with melanoma progression was also reviewed recently [52,62,568]. The above clinical finding is supported by previous experimental data on cultured melanoma cells showing attenuation of inhibitory effects of vitamin D derivatives on melanomas showing decreased or absent VDR [439,489,569,570].

With respect to non-classical vitamin D receptors, RORα and RORγ, their expression negatively correlated with the progression of pigmented tumors [408]. Both nuclear and cytoplasmic RORα and RORγ decreased with progression lesions from nevi to primary melanomas to melanoma metastases. Similarly, for VDR, RORα and RORγ expression decreased in melanomas with active melanogenesis. Melanomas at higher advancement (Clark levels III-V and Breslow thickness > 2 mm) presented the lowest levels of both nuclear and cytoplasmic expression of RORα and RORγ. Furthermore, RORα and RORγ expression correlated to melanoma prognostic markers and was lower in melanomas with higher proliferation rate, presence of ulceration, absent and non-brisk TILs, and nodular histological type [408]. The expression of enzymes involved in metabolizing vitamin D, CYP27B1 and CYP24A1, also diminished [407,438] (Figure 4). CYP27B1 expression decreased with the progression of melanocytic tumors in the following order: nevi—melanoma—melanoma metastases. Melanomas at the radial growth phase showed comparable CYP27B1 expression to nevi, and superficial spreading melanomas showed comparable CYP27B1 expression to nodular melanomas. The melanomas with the presence of other aggressive phenotype markers (higher proliferation index, vertical growth phase, Clark levels III–V and Breslow thickness ≥ 2.1 mm, developing the metastases) had lower CYP27B1 expression. The relation of CYP27B1 expression to melanoma prognosis is also reflected by shorter overall survival and disease-free survival in patients with reduced CYP27B1. Similarly to VDR, CYP27B1 expression was also lower in highly pigmented melanomas than in amelanotic or slightly pigmented melanomas [407]. Similarly, CYP24A1 expression decreased with the progression of melanocytic tumor from nevi to primary melanomas to metastases. Melanomas stratified according to their Breslow thickness, Clark level, pT, and overall stage also showed the lowest CYP24A1 level in the most advanced melanomas (Clark levels IV-V, Breslow thickness > 2 mm, pT3-4, stage 3–4) [438]. In addition, recent studies have indicated that the expression of VDR, RORα, and RORγ in melanomas is related to hypoxia and/or HIF1-α activity, and also affected FoxP3 expression in metastatic melanoma [57]. Accordingly, it was proposed that hypoxia can affect tumor behavior by changing NRs expression and downstream immune responses [52,57].

It is worth noting that not only the vitamin D endocrine system is disturbed in cutaneous melanomas but the impairment of this system was also found in uveal melanomas [55]. Uveal melanoma cells were characterized by the lowest expression of VDR, RORα, CYP24A1, and CYP27B1 than the other uveal cells. 

#### 5.2.2. Molecular Studies

The *VDR* gene contains 11 exons and is located on chromosome 12q13.11 [571]. Single nucleotide polymorphisms (SNPs) in the *VDR* gene can affect the expression and/or function of the VDR protein, potentially influencing the incidence and natural history of melanoma [56,122,124,429,554,572,573,574,575,576,577]. Among over 600 single nucleotide polymorphisms (SNPs) in the *VDR* gene FokI (C/T-rs2228570 (previous name rs10735810), TaqI (rs731236), BsmI (rs1544410), and ApaI (rs7975232) have been most commonly analyzed for their relationship with melanoma, with Cdx2 (rs11568820), EcoRV (rs4516035), and BglI (rs739837) studied to a lower extent. The FokI SNP is located on exon 2 of the *VDR* gene, which generates an alternative start codon 10 base pairs upstream from the expected start codon, leading to production of longer and less active VDR proteins [56,578]. The TaqI SNP is at codon 352 of exon 9 of the VDR gene and creates a silent codon change of ATT to ATC, which both code for isoleucine [56,579]. The BsmI SNP generates a silent mutation, is located in intron 8 of the *VDR* gene, and can affect gene expression and mRNA stability [56,578,579]. The Cdx2 and SNPs are in the promoter region of the *VDR* gene, respectively [56,579] with the latter suspected to play a role in the anticancer immune responses [580]. The BglI SNP is near the stop codon in exon 9 of the *VDR* gene [56].

Although individual studies carried out on small cohorts of patients reported controversial results, there is a consensus that there is a relationship between VDR gene polymorphisms and melanoma risk and progression as relates to selected SNPs that would affect vitamin D signaling [53,56,62,568,572]. For example, VDR variants FokI, ApaI, and BsmI may influence the susceptibility to developing melanoma [53,581]. However, no significant association between melanoma risk and VDR variants TaqI (rs731236), A-1012G (rs4516035), Cdx2 (rs11568820), and BglI (rs739837) was found [53]. Also, there is no clear relationship between SNPs on vitamin D binding protein (VDBP) [582,583], CYP27B1, and CYP24A1 [576] and melanoma. In other cancers, selected SNPs for CYP24A1 may reduce the risk or cancer presentation [584,585,586]. It must be noted that in all gene polymorphism analyses for melanoma patients, novel pathways of vitamin D activation or their receptor targets were not taken into consideration. Furthermore, analysis of 703 primary melanoma transcriptomes and 365 samples of metastatic melanoma from The Cancer Genome Atlas (TCGA) showed that VDR expression was independently protective for melanoma-related death in both primary and metastatic disease, and high tumor VDR expression was associated with upregulation of pathways mediating antitumor immunity and corresponding with higher imputed immune cell scores and histologically detected tumor-infiltrating lymphocytes [54]. In addition, increased expression of the VDR correlated with downregulation of proliferative pathways, notably Wnt/β-catenin signaling [54]. These were confirmed by in vitro inhibition of human melanoma growth associated with inhibition of Wnt signaling, and reduced ability to form lung metastases by B16 melanoma overexpressing VDR was again associated with decreased Wnt/β-catenin signaling [54].

### 5.3. Experimental Data on Anti-Melanoma Activity of Vitamin D

#### 5.3.1. In Vitro Studies

Colston et al. [284] was the first to show an inhibitory effect of 1,25(OH)_2_D3 on melanoma cell proliferation accompanied by the expression of the VDR. The earliest reports also included a description of inhibitory effects of not only 1,25(OH)_2_D3, but also its metabolites on melanoma growth [587] and the capability of melanoma cells to transform 25(OH)D3 to 1,25(OH)2D3 and 24,25(OH)2D3 [444]. Since then, there have been several reports showing anti-melanoma activity of active forms of vitamin D (reviewed in [51,63,555,588,589]). The majority of the studies were performed on human melanoma cell lines and their growth inhibition by 1,25(OH)_2_D3 was dependent on VDR expression [51,439,441,569,570,588,590]. Regarding rodent melanoma cell lines, vitamin D3 hydroxyderivatives showed anti-melanoma activity in a hamster melanoma line [410,441] and by some authors in B16 melanoma [410,591]. Other studies have shown the lack of an effect of vitamin D hydroxyderivatives on B16 [592] and S91 [512] melanoma cells. Relating to murine B16 melanoma, the reported differences in biological effects of vitamin D derivatives could be secondary to test conditions or simply use of different sublines of B16 melanoma [588]. A variety of vitamin D3 and D2 derivatives were tested including classical forms, chemically generated low calcemic forms, and CYP11A1-derived non-calcemic or low calcemic variants, all of them showing anti-melanoma activities on cultured melanoma cells [51,63,188,285,410,411,439,441,443,449,461,483,484,487,489,533,569,588,593,594,595,596,597]. In some melanomas, 1,25(OH)_2_D3 or its synthetic analogs showed proapoptotic effects, while in others there was a lack of such effect [188,598]. Also, our studies indicated the lack of pro-apoptotic effects by D3 hydroxyderivatives. Interestingly, anti-melanoma activities in vitro have been demonstrated for vitamin D derivatives with a shortened or absent side chain (pregnacalciferols (pD) and pregnalumisterols (pL)) [410,443,449,469,473,474,484,588], which show reduced calcemic activity [599]. In addition, recently discovered lumisterol hydroxy metabolites have shown anti-melanoma activity in cell culture [464,465,496,600]. Of note, these compounds are non-calcemic.

Vitamin D derivatives not only have a direct anti-melanoma effect, but also can sensitize melanoma cells to different types of therapy [592,594,595,596,601]. For example, 1,25(OH)_2_D3 and 25(OH)D3 can sensitize melanoma towards proton beam therapy [601]. In human melanomas in culture, they modulate anticancer properties of cisplatin and dacarbazine [594], and enhance anticancer properties of cediranib, a VEGFR inhibitor [596], and of vemurafenib [595]. Concerning the relationship between melanogenesis and vitamin D, it has been shown that the amelanotic phenotype makes melanoma cells more sensitive to active forms of vitamin D in comparison to melanotic ones [411]. Others have shown more complex relationships, with moderate pigmentation sensitizing rodent melanoma cells towards vitamin D analogs with altered expression of key genes involved in vitamin D signaling. This was opposite to the effect on heavily pigmented cells in which strong melanization inhibited ligand-induced translocation of VDR to the nucleus [410]. It should be noted that clinico-pathological data have demonstrated that high melanization inhibits the expression of VDR and CYP27B1 [52,406,407].

Novel CYP11A1-derived metabolites of vitamin D, which are detectable in the human body and show low or no toxicity in comparison to 1,25(OH)_2_D3 [72,445], deserve special attention. Specifically, 20(OH)D3 and 20(OH)D2 and their hydroxy metabolites with or without an OH at C1α have shown anti-melanoma activities, including inhibition of cell proliferation, colony formation in monolayer and soft agar, migration capability, and spheroid formation [63,285,410,411,441,449,487,489]. In general, mono-, di- and tri-hydroxy metabolites of 20(OH)D3, (20,23(OH)_2_D3, 20,24(OH)_2_D3, 20,25(OH)_2_D3, 20,26(OH)_2_D3, and 17,20,23(OH)_3_D3), including their C1α hydroxyderivatives of the dihydroxy forms, were more potent than the 20(OH)D3/2 precursor molecule. For example, products of CYP24A1 hydroxylation (20,24(OH)_2_D3, 20,25(OH)_2_D3, and 20,26(OH)_2_D3) were more potent at inhibiting melanoma growth than their precursor 20(OH)D3 [461]. Also, their C1α hydroxyderivatives were more potent than 20(OH)D3 [452].

For the human WM164 melanoma these effects were partially dependent on the VDR as demonstrated in experiments with knocked down (KD) VDR vs. mock transfected cells (wild type: WT) [489]. To get a better insight into the regulation of melanoma transcriptosome, we isolated RNA from these cells treated for 24 h with 1,25(OH)_2_D3, 1,20(OH)_2_D3, 25(OH)D3, 20(OH)D3, or 0.1% ethanol (control) and submitted these samples for RNA sequencing by BGI Americas Corporation Service (Cambridge, MA, USA), as previously described [493,494]. The resulting raw data were deposited at GEO (GSE248660). The initial analysis showed significant changes in the transcriptosome of the VDR-KD vs. VDR-WT (Appendix A) with 155 or 128 genes expressed only in KD vs. WT cells and with 739 or 562 genes upregulated or downregulated using a cut of >2 (Appendix A). This produced significant differences in expression of major canonical pathways, diseases, biofunctions, and upstream regulators, as analyzed by Ingenuity Pathway Analysis (IPA) (Appendix A). Heat maps and Venn diagrams show significant differences in gene expression and distinct and overlapping genes affected by treatments with 1,25(OH)_2_D3, 1,20(OH)_2_D3, 25(OH)D3, and 20(OH)D3 (Appendix A). These analyses further indicate differential activity of different vitamin D3 hydroxyderivatives on a number of different receptors in addition to the VDR, as discussed in the next Section on signal transduction.

In addition, CYP11A1-derived lumisterol hydroxy metabolites inhibited melanoma proliferation [464]. Finally, secosteroids with a shortened or absent side chain, pL compounds, 7DHP, and other Δ7-steroidal derivatives, have shown anticancer effects on human, hamster, and murine melanoma cell lines [410,443,449,469,473,474,484,594,602].

#### 5.3.2. In Vivo Studies

To test the hypothesis that vitamin D3 could be an effective anti-melanoma agent, several in vivo studies were conducted. Specifically, 1,25(OH)_2_D3 inhibited the metastatic process in a B16 melanoma model, while having no effect on tumor growth in a subcutaneous location [591]. 1,25(OH)_2_D3 at pharmacological doses inhibited growth of VDR positive human melanoma xenografts in vivo, while being well tolerated when a low calcium diet was used [603]. In the Tyr-Tag transgenic mouse, 1α(OH)D2 inhibited pigmented ocular tumor growth at moderate drug levels with relatively low mortality [604]. In fact, the total body score was higher in the treatment group compared to control group (2.8 vs. 2.55). Most recently it has been shown that topical application of calcipotriol plus imiquimod (IMQ) inhibited growth of B16 melanoma (clone F10) [605]. Overexpression of VDR in B16 melanomas lead to decreased ability to form lung metastases [54]. Similarly, protective functions of VDR in neonatal mouse melanocytes in vivo against UVB-induced DNA damage was demonstrated [281]. Moreover, loss of keratinocytic RXRα, a dimeric partner of VDR, combined with activated CDK4 or oncogenic NRAS generated UVB-induced melanomas [606]. DMBA/TPA topical treatment of *VDR*-null mice increased formation of dermal melanocytic nevi in comparison to controls [607]. The above results prompted us to hypothesize that low calcemic vitamin D metabolites such as 20(OH)D3 could be more effective and less toxic to prevent melanomagenesis. Along that line, 20(OH)D3 at 30 μg/day inhibited the growth of human melanoma implanted into NOD.Cg-*Prkdc^scid^ Il2rg^tm1Wjl^*/SzJ (NSG) mice without showing any signs of toxicity [285]. Furthermore, we have seen an enhancement of the vemurafenib anti-melanoma effect by 20(OH)D3 in human WM164 melanoma implanted into the nude mice (Podgorska et al., in preparation). Also, 20(OH)D3 can inhibit melanoma progression in *BRAF^V600E^|PTEN-null* mice in vivo (Indra et al., in preparation). It remains to be determined whether the anticancer effects of the vitamin D hydroxy derivatives are mediated by VDR and/or alternative receptor(s) in vivo, in addition to elucidating the mechanism of action of those novel metabolites in the melanoma microenvironment. 

The demonstrations of anti-melanoma activity by vitamin D compounds in cell culture, in animal studies as well as those inferred from data in human subjects, point to a role for vitamin D compounds as preventive agents to reduce UV-induced DNA damage and as adjuvant therapy for melanoma, in combination with other approaches, including checkpoint inhibitors and mRNA therapy to neoantigens. In moderate doses, vitamin D compounds are not toxic and may enhance immune system removal of neoplastic cells. The above studies mandate more extensive studies on inhibition of melanoma growth in vivo by different D3 derivatives or testing whether a D3-deficient or -sufficient diet would have an effect on melanoma growth and metastatic capability in animal models of melanoma.

#### 5.3.3. Mechanism of Action

The proposed mechanistic outcome of D3 and L3 activation on melanoma development and therapy through activation of classical, VDR, and non-canonical (AhR, RORs, LXRs, PPARγ) nuclear receptors is presented in Figure 5. The rationale for chosen receptor targets and signaling pathways is presented in the preceding subchapters. Below we provide additional considerations not discussed above.

With respect to melanomagenesis, there is evidence that incomplete repair of DNA damage generated by UVA and UVB contributes to melanoma development, especially on skin that is exposed to the sun [80,118,254,255,259,260,271,272,278,608,609,610]. When DNA damage is inadequately repaired, it may lead to mutations or amplifications of genes such as BRAF, Kit, and cyclin D1, which are involved in a variety of survival and growth pathways. Melanocytes differ from keratinocytes in that they are considerably less proliferative and appear to have less DNA repair capacity, but tend also to be apoptosis resistant [608,610]. Chromatin remodeling is essential to DNA repair and 1,25(OH)_2_D3 is known to alter chromatin by epigenetic mechanisms [611,612]. The VDR/RXR dimer recruits histone steroid receptor coactivators 1 and 2 as well as acetyltransferases, which in turn open up chromatin [613]. 1,25(OH)_2_D3 has been shown to increase DNA repair enzymes XPC and XPA in human skin explants [467]. It should be noted that the protective role of the VDR in UVR-induced skin cancers, including melanoma, connected to stimulation of anti-oxidative and DNA repair responses is widely recognized [62,271,429,533,568,614,615,616,617,618], as discussed above.

1,25(OH)_2_D3 was shown to increase caspase activity and cell viability in those human melanoma cell lines which expressed both tumor suppressors PTEN and the VDR [533]. In some melanoma cell lines, treatment with 1,25(OH)_2_D3 resulted in significant increases in PTEN, along with reduced phosphorylation of AKT and downstream pathways [533]. Mutation or loss of PTEN has been associated with melanoma development [534]. DNA repair is likely facilitated by further increases in p53 in melanocytes with 1,25(OH)_2_D3 after UV [525], and likely by many of the naturally occurring skin metabolites of vitamin D, lumisterol, and tachysterol [64,65,66,466,467,490,526,532]. This increase in p53 probably contributes to enhanced apoptosis of badly damaged melanocytes and to increases in XPC. Also to be noted is that protein disulfide-isomerase A3 (PDIA3; 1,25D3-MARRS; ERp57) and non-genomic A-VDR were described as “membrane vitamin D receptors” responsible for rapid nongenomic photoprotective responses [490,501,529] (Figure 3), which could act in addition to nuclear receptor signaling, as presented in Figure 5.

Energy is required for DNA repair, but energy generation by skin cells is reduced after UVR [619,620], in part due to damage to mitochondria [621] and due to increased poly(ADP-ribose) polymerases (PARPs), especially PARP-1, which inhibits glycolysis [622]. Treatment of skin cells with 1,25(OH)_2_D3 improves mitochondrial repair and increases glycolysis [279]. It is likely that these processes enhance the otherwise reduced DNA repair in melanocytes. The mechanism(s) by which active forms of vitamin D3 can affect mitochondrial functions remain an enigma. However, there are reports demonstrating that 1,25(OH)_2_D3 can interact with mitochondrial potassium channels [497,498], indicating its direct receptor independent action on mitochondria. In addition, VDR involvement in regulation of mitochondrial function is indicated by studies showing VDR localization in mitochondria and that VDR could affect mitochondrial functions directly [623,624,625,626,627]. We also have found partial co-localization of the VDR and CYP11A1 in mitochondria of keratinocytes (Figure 3 in [64]). Since VDR is lacking an N-terminal mitochondrial import sequence, it appears that the receptor is transported via a permeability transition pore (PTP)-dependent pathway [624].

We have confirmed the above findings using image-flow cytometry by showing that 1,20(OH)_2_D3 and 20,23(OH)_2_D3 induced translocation of the VDR to the mitochondria that was comparable to the receptor translocation to the nucleus (Figure 6). For these assays, we used HaCaT keratinocytes, the same cell line reported by Silvagno et al. [624,625]. Following treatment with 1,20(OH)_2_D3 or 20,23(OH)_2_D3, significantly greater levels of VDR were observed within mitochondria and nuclei with increased mitochondria/cytoplasm and nuclei/cytoplasm ratios, indicating that these compounds induced translocation of the receptor to these organelles (Figure 6). Since AhR and RORγ serve as alternative nuclear receptors for CYP11A1-derived vitamin D3 hydroxyderivatives, we also tested the effect of 1,20(OH)_2_D3 and 20,23(OH)_2_D3 on induced expression of these receptors in mitochondria in comparison to nuclei (Figure 6). Similarly, image-flow cytometry showed 1,20(OH)_2_D3 and 20,23(OH)_2_D3 induced translocation of AhR and RORγ to mitochondria and nuclei, with 20,23(OH)_2_D3 having greater effect than 1,20(OH)_2_D3 (Figure 6). These results not only support previous studies on VDR expression in mitochondria [624,625] and a modulatory effect of vitamin D3 hydroxyderivatives on mitochondrial functions in keratinocytes [64,279,497], but also indicate that they can induce the translocation of alternative nuclear receptors with efficacy dependent on the position of hydroxy-groups in the vitamin D3 structure. The documentation presented in Figure 6 is important, since it indicates ligand-induced translocation to the mitochondria and nucleus, explaining the reported effects of vitamin D analogs on mitochondrial functions.

With respect to the transcriptional regulators listed in Figure 5, regulation of p53 activity by D3 hydroxyderivatives has been mentioned above. Additionally, positive regulation of p53 by the VDR with downstream photoprotective and anti-melanoma effects were discussed by others [271,617,628]. The role of p53 as a target for anti-melanomagenesis and melanoma therapy is also indicated by many studies [10,97,98,165,255,334,629,630]. Similarly, the loss of keratinocytic RXRα (dimer partner for VDR) combined with activated CDK4 or oncogenic NRAS generated UVB-induced melanomas via loss of p53 and PTEN [606]. Recent studies have demonstrated that 1,25(OH)_2_D3 through activation of VDR stimulates PTEN and inhibits AKT with a net inhibitory effect on melanoma cells [533]. PTEN is a good pharmacological target in melanoma therapy [6,134,534,631], which would include inhibition of the PI3K-AKT signaling pathway [533,534]. It has also been reported that ligand-induced VDR signaling can inhibit Wnt/β-catenin-mediated progression of melanoma and enhance antitumor immunity [54]. Thus, vitamin D hydroxyderivatives can also target the Wnt/β-catenin pathway, in a similar manner to that described for squamous cell carcinomas [280,616,632]. The role of NFκB in melanoma development and resistance to therapy through generation of a pro-inflammatory environment and through anti-apoptotic activity has long been recognized [151,629,633,634,635,636,637,638]. Vitamin D3 hydroxyderivatives are not only capable of inhibiting NFκB activity in epidermal keratinocytes, but also in melanoma cells, which correlates with anticancer activity [411]. Interestingly, a recent study has shown that the precursor to vitamin D3, 7DHC, can suppress melanoma cell proliferation and invasion via inhibition of AKT1/NFκB signaling [639].

The role of IL-17 signaling in melanoma development and therapy is still controversial with antitumor and protumor effects [640]. Specifically, studies on B16 melanoma demonstrated that increased IL-17 signaling correlates positively with formation of melanoma and their growth in mice, while its inhibition has an anti-melanoma effect [641,642,643]. Similar effects were shown in human melanoma cell lines with increased IL-17 expression in melanoma in comparison to peritumoral tissue [644]. Similar trends of increased expression of IL-17 in nevi vs. normal skin, with increased staining in melanomas vs. nevi with increased intensity in invasive tumors vs. melanoma in situ was observed by others [645]. On the other hand, increased IL-17 signaling and serum levels of IL-17 correlated positively with positive outcome of dual CTLA-4 and PD-1 checkpoint inhibition in melanoma and improved OS of the patients [646]. Furthermore, other investigators using animal models of melanoma have shown that IL-17A-deficient mice are susceptible to spontaneous melanoma development [647]. However, the regulation and role of IL-17 are complex, including regulation of IL-17 in melanoma by CYP11A1-derived vitamin D3 hydroxyderivatives and transcriptional regulation of IL-17 by RORγ [648,649].

Similar dual roles of NRF2 in protection against melanomagenesis and promotion of tumor growth were discussed recently [76,650]. Similar dual anti- and pro-cancerogenic functions of NRF2 have been described for other cancers, with the latter suggesting hijacking of NRF2 by cancer cells for their survival [651]. Thus, NRF2 through induction of antioxidant and detoxification responses will attenuate malignant transformation of melanocytes and perhaps initiate tumor progression [76,650,652], while it will stimulate progression to aggressive forms and increase resistance to therapy [76,650,653,654,655]. However, NRF2 can exert tumor suppressor activity in a context dependent fashion [656]. The latter indicates that NRF2 may not only play a role in melanoma prevention but also in therapy, in a context dependent fashion. Activation of AhR can stimulate NRF2 [657,658] with implications in cancers [658,659] and UVB-induced skin damage [660]. We hypothesize that NRF2 activity can be regulated by vitamin D3 hydroxyderivatives to reduce UVB-induced damage in melanocytes and inhibit their transformation toward a melanoma phenotype. However, we also note a possible dual role of AhR in melanoma natural history, including its protective and protumor roles. For example, AhR can directly promote resistance to BRAF inhibitors [231,661] and vemurafenib can act as an AhR antagonist [662]. Taking into consideration the wide range of AhR ligands with sometimes conflicting phenotypic actions [650,659,663,664,665,666,667], we believe that therapeutic activation of AhR will be ligand-dependent. In this context, studies on the therapeutic role of vitamin D and related molecules through targeting AhR are worth pursuing.

The above considerations indicate vitamin D and lumisterol hydroxyderivatives can exert their chemopreventive or therapeutic activities against melanoma through targeting specific nuclear receptors with activation of downstream signaling pathways. We predict that such receptor activation will be defined by the structure of the compound and number and location of hydroxyl group in their structure. For example, vitamin D derivatives containing an OH on C1α exhibit increased selectivity towards VDR [448,481,482,484,485], while compounds without C1α(OH) act predominantly on alternative nuclear receptors [409,468,483,486,491,495] and for which selectivity is defined by the number and position of OH groups on the side chain (Jetten, Slominski et al., in preparation). 

### 5.4. Clinical Data on Anti-Melanoma Actvity

Numerous natural compounds exhibit anti-melanoma activity targeting either tumor formation or progression. Some of these nutraceuticals are used alone in the treatment of melanoma, while others, such as vitamin D, are used in combination with other anti-melanoma modalities [668]. The anti-melanoma potential of vitamin D has been studied in clinical trials. Liang at al. conducted a clinical trial on women which showed that low vitamin D levels in serum puts them at higher risk for non-melanoma skin cancer development [669]. In addition, Vincenti et al. observed that low vitamin D levels put people of Northern Italy at risk for melanoma occurrence [670]. Vitamin D exhibits a plethora of biological activities and interferes with different signaling mechanisms, therefore it is extensively used in treatment of different diseases, including melanoma. In addition to many studies on human and murine melanoma cells, vitamin D has been used in human clinical trials on patients with different diseases, including melanoma [671]. Since this is a natural compound and non-toxic, it is well tolerated even at high doses alone or in combination with other therapeutic agents. The therapeutic efficacy of vitamin D was tested on melanoma cells [411] and was successfully used in hindering melanoma in a mouse model [605]. High doses of vitamin D were shown to be beneficial in decreasing the risk of melanoma recurrence after treatment, as seen in a clinical trial on melanoma patients [59,672]. In a phase II trial on melanoma, performed in Australia, a high dose of oral vitamin D was included as an adjuvant therapy in patients with melanoma, which improved the overall survival rate [672]. In another study with anti PD-1 therapy against melanoma, supplementation with vitamin D improved treatment outcomes [59].

Numerous clinical trials for melanoma treatment have been completed, yet many more are still ongoing. A search on www.clinicaltrials.gov (accessed in 3 May 2024) using the key words “human” and “melanoma” indicated 124 ongoing studies, while the key word “melanoma” resulted in 3266 studies worldwide, out of which 1432 are already completed. Using key words “vitamin D” and “melanoma”, the search indicated 10 ongoing studies. Five studies using vitamin D supplementation in melanoma treatment are already completed, one is still ongoing, while the others were terminated. The Vitamin D Supplementation in Cutaneous Malignant Melanoma Outcome (ViDMe) (NCT01748448) study started in 2012 and was completed in 2023 in Belgium. It was designed to assess the potential of vitamin D supplementation in melanoma relapse after surgery in a broad population from young adults to older adults. The initial results, published in 2017, showed that vitamin D supplementation may reduce the relapse of cutaneous malignant melanoma [673]. In addition, it was reported that regular use of vitamin D supplement was associated with fewer melanoma cases compared to non-use [60].

The four-year-long “Co-stimulatory Markers and Vitamin D Status in Anti-PD1 Treated Melanoma Patients (NCT03197636)” trial in Denmark combined PDL therapy for advanced melanoma in combination of vitamin D. This trial ended in 2022 and no study results are available yet. A trial in Illinois on 20 participants with metastatic melanoma, initiated in 2005, was designed to determine the best dose of calcitriol for the treatment of melanoma in addition to temozolomide. Hypothetically, calcitriol should sensitize tumor cells, thus making them more prone to the influence of temozolomide. The study results are not published yet.

Although low vitamin D status around the time of diagnosis is associated with worse prognostic indicators and worse survival in patients with melanoma, whether supplemental vitamin D as adjuvant treatment would improve melanoma prognosis is unclear. In women with a history of non-melanoma skin cancer, as determined by the Women’s Health Initiative Trial, those who received calcium (1000 mg) and vitamin D (400 IU/day) had a reduced risk of melanoma, although this was not seen in the whole cohort [674]. One pilot clinical trial was established mainly to examine the safety (particularly hypercalcemia) of large intermittent doses of vitamin D in melanoma patients soon after diagnosis [672] (Australia and New Zealand Clinical Trials Registry (ANZCTR) ACTRN12609000351213). From more recent studies, the dosing regimen was probably not ideal for optimal vitamin D repletion [675,676]. No safely issues were reported, but the trial was not powered for a clinical outcome (R. Saw, personal communication). Adjuvant vitamin D supplementation of 100,000 IU/50 days after resection of a stage II melanoma used in another trial reported that the primary determinant of disease-free survival was Breslow thickness (<3 mm vs. ≥3 mm) with no significant effect of vitamin D supplementation [677]. This study also reported the so far unexplained observation that blood 25(OH)D concentrations increased to a significantly lesser extent in patients with a Breslow score ≥ 3mm [677].

Given the great improvements in survival after melanoma with immune therapy, it is not surprising that there appears to be a limited number of other trials registered to examine whether vitamin D supplementation improves melanoma outcomes. One such trial established to examine this question [673] (Clinical Trial.gov, NCT01748448, 5 December 2012) has not reported to date. It seems likely that future trials may examine the role of vitamin D supplementation as adjuvant treatment, rather than as a primary approach. Also, higher doses of vitamin D may be required since vitamin D3 supplementation at 10,000 IU/d produced genomic alterations several fold higher than 4000 IU/d, even without further changes in PTH levels [678]. Moreover, parenteral routes of delivery could be considered, since the majority of orally delivered vitamin D3 will be hydroxylated to 25(OH)D3 in the liver, attenuating its availability to CYP11A1, as discussed in [679,680]. Also, novel CYP11A1-derived vitamin D3 hydroxyderivatives are awaiting clinical trials, since they are non-toxic and non-calcemic at suprapharmacological doses, being detectable in natural products including honey [681,682]. This is in addition to their anti-melanoma and anticancer activities described above.

### 5.5. Bioinformatics Considerations of Melanoma Diagnosis and Treatment

#### 5.5.1. An Overview

In the past twenty years, bioinformatics has been playing a key role in the treatment and diagnosis of melanoma. An enormous amount of data has already been published or deposited in different databases such as the Gene Expression Omnibus (GEO) developed by the NIH [683], or the cBioPortal, a cancer genomics data repository developed by the Memorial Sloan Kettering Cancer Center [683,684]. The GEO database itself contains 13,905 entries related to tissues and 216 entries related to strains from Homo sapiens. Altogether, for Homo sapiens, it includes 38 datasets, 29 series, and 49,571 samples supported by 20 platforms. The study types are diverse (around 26), encompassing various profiling techniques such as expression profiling by array, high throughput sequencing, SNP array, etc. Similarly, cBioPortal and other databases contain data of immense value to researchers to help understand melanoma better. These omics data with comprehensive bioinformatics and computational biology approaches are in agreement with classical pathological and clinical predictors, as well as overall survival (OS) and disease-free survival (DF), and should aid in defining signaling pathways and crosstalk between multiple pathways driving tumor progression, as well as resistance to therapy or inhibiting it. Many examples of signaling involved in tumor progression and resistance to therapy were discussed recently [50,82,183,229,230,685] including the capability of a tumor to affect its local environment and systemic responses [82]. One crucial pathway that has been shown to inhibit the malignant behavior of melanoma is vitamin D signaling, different aspects of which could become lost during the progression of the disease [51,52,62,63]. The mining of the data could require artificial intelligence (AI) to develop a unified staging model encompassing classical and molecular parameters, forming a basis for in silico modeling and dynamic simulation tests. Computational and immunogenomic approaches for cancer immunotherapies are already being discussed for different types of cancer [686,687,688,689]. AI tools are already used to assist in prognostication or the prediction of treatment responses of melanoma patients to immunotherapy [690] and in general diagnosis and therapeutic predictions [30,343,688,690,691,692,693,694,695]. Furthermore, recent advances in integrating spatial transcriptomics with histology promises to revolutionize pathological diagnosis of tumors [696], including melanoma (see below).

#### 5.5.2. Data-Centric Research in Melanoma: Enhancing Outcomes through AI and Data Science

Data-centric research in melanoma focuses on systematically collecting, analyzing, and using diverse data types to enhance the understanding and management of melanoma. This approach prioritizes data as a critical asset, strategically generating insights and informing decisions in medical research and clinical settings (Figure 7). Melanoma’s complexity, marked by varied genetic mutations and patient treatment responses, highlights the need for data-centric research. This method aids in the development of personalized medicine, enabling treatment plans tailored to individual patient profiles, which may lead to significantly improved outcomes [693,697,698].

Accordingly, the following types of data for melanoma can be gathered: (1) Clinical data that includes patient records, treatments, and outcomes, offering real-world insights into the efficacy of medical interventions [699]; (2) Genomic/genetic data [700,701,702]; (3) Imaging data, including dermatoscopic and histological images which are essential for early and accurate melanoma diagnosis [703,704]; (4) Environmental data such as UV exposure, lifestyle, and geographic factors influencing melanoma risk and progression [705,706]. As presented in Figure 7, AI significantly enhances data-centric research capabilities in melanoma by: (1) Diagnosis where AI algorithms analyze complex imaging data to differentiate malignant from benign lesions with high accuracy [703,707]; (2) Prognosis and treatment personalization where predictive models integrate genomic and clinical data to forecast disease progression and customize treatment strategies [708,709,710,711,712]; (3) Epidemiological insights where AI analyzes environmental and behavioral data to identify risk factors and inform public health strategies [713].

Integrating AI into melanoma research presents several challenges: (1) Data privacy and security with protecting sensitive patient data in line with regulations such as GDPR and HIPAA; (2) Data quality and integration to manage data inconsistencies, which can impact AI model accuracy [714]; (3) Algorithmic bias, and addressing biases in AI applications to ensure effectiveness across diverse populations; (4) Ethical considerations in the impact of AI on clinical decision-making and the implications of its predictions.

Therefore, we conclude that AI and data science are crucial in advancing melanoma research and clinical practice, refining prevention strategies, improving diagnostic accuracy, and personalizing treatments. Integrating these technologies allows for the development of more effective and tailored approaches to combat melanoma, addressing treatment effectiveness and adaptation to individual patient needs. Ongoing collaboration among healthcare providers, researchers, patients, and regulators is vital to overcome challenges and maximize the benefits of AI in melanoma management [703].

The challenging question is how to perform data-centric research in melanoma. Figure 8 provides a structured overview of a data-driven diagnostic system for melanoma enhanced by AI. It illustrates the placeholders of systematic collection of critical health data, the development of predictive AI models, and the use of these models to make timely and accurate predictions.

These include data collection stages: (1) Initial Clinical Data (D1) where clinicians collect initial data, including patient phenotype, family history, environmental exposures, and nutritional factors, which may raise preliminary diagnostic questions or indicate a need for further testing; (2) Advanced Diagnostic Data (D2) where subsequent data collection includes detailed analyses like omics, pathology imaging, and other relevant pathology data to refine the diagnostic process.

The AI model development includes: (1) Predictive Diagnostics (M1), where an AI model is designed to predict initial diagnoses, minimizing early-stage errors; (2) Disease-Specific Prediction (M2), where an AI model focuses on accurate diagnosis of specific conditions like melanoma, offering precise predictions; (3) Comprehensive Reporting (M3) where an AI report generator compiles findings into an intelligible report, summarizing the AI’s analytical process.

The Predictive Outcomes are as follows: (1) Initial Diagnosis and Prognosis (U1), where AI predicts initial diagnoses and prognoses to reduce false negatives and enable early intervention; (2) Accurate, Timely Treatment Recommendations (U2), where AI recommends precise treatment plans with fast execution time for timely delivery. 

Thus, integrating AI in the medical diagnostic process enhances clinicians’ ability to provide accurate diagnoses and prognoses. It contributes to more informed decision-making for treatment plans. By reducing diagnostic times, AI ensures that patients receive timely, effective treatments, optimizing outcomes in clinical practice. We plan to apply these considerations in exploring utility of vitamin D signaling in diagnosis, prognostication and/or primary or adjuvant therapy in melanoma.

## 6. Concluding Remarks and Perspective

Cutaneous malignant melanoma still represents a significant clinical problem despite recent advances in diagnosis and therapy due to its resistance to therapy and unpredictable behavior. Also, despite significant progress in diagnosis, classical and molecular classification of the disease and defining regulatory mechanisms underlying the process of melanomagenesis in relation to environmental, constitutive, and genetic factors, as well as principles of melanoma progression, our knowledge of mechanisms regulating these processes is limited. This is exemplified by our inability to cure the disease, unless the melanoma is localized and surgically removed. With respect to advanced melanomas, the ability of the tumor to regulate both its environment and a global homeostasis influencing the systemic neuro-immuno-endocrine regulators which would promote melanoma growth is worth mentioning. Furthermore, although environmental factors such as UVR and genetic/constitutive background for melanoma susceptibility are well defined and acknowledged by the public, the incidence rate of melanoma is continuously increasing. This cannot be solely explained by over-diagnosing dermatopathology interpretations, because the mortality rate, instead of decreasing, as would be expected from improved early detection, is slightly increasing. This indicates that our preventive measures are not only insufficient or defective, but we may also be missing some unrecognized factors involved in a complex process of melanomagenesis. Therefore, to better understand the etiology and the process of melanomagenesis and melanoma progression, necessary to optimize preventive and therapeutic strategies, we have to use comprehensive strategies to mine deposited data including traditional pathology and new omics data as well as existing clinical and experimental knowledge on this disease. This can best be approached with the use of computational biology and application of AI tools to solve any emerging problem because of the complexity of the process and vast information already available on melanoma that is exponentially growing.

UVB radiation, while being a recognized environmental carcinogen, is required for cutaneous production of vitamin D3, which can be enzymatically activated to biologically active hydroxyderivatives. These have pleiotropic effects including anti-melanoma activities and protective or reparative effects against oxidative stress and DNA damage induced by solar radiation. Moreover, evidence is accumulating that vitamin D deficiency, defined by ≤20 ng/mL (50 nmol/L) of 25(OH)D3, and defects in vitamin D signaling involving VDR and CYP27B1 in the canonical pathway affect the probability of melanoma development and natural history of the disease, including OST, DFST, and response to therapy. These are further supported by context-dependent anti-melanoma activities of vitamin D3 derivatives in experimental models. Recent advances in vitamin D biology also identified alternative CYP11A1-dependent pathways of vitamin D activation producing large number of hydroxyderivatives that show anticancer (including anti-melanoma) and photoprotective properties, while acting as biased agonists on the VDR and/or as (inverse) agonists on several other nuclear receptors or by inducing nuclear receptor independent signaling. Furthermore, UVR-produced lumisterol and tachysterol, previously considered as biologically inactive, can be activated enzymatically to several products also showing anticancer, anti-melanoma, and radioprotective properties with action on selected nuclear receptors or receptor-independent mechanisms of action. Thus, a new UVB-induced ecosystem has been uncovered (Figure 2, Figure 3 and Figure 5), which can have preventive or attenuating functions in melanomagenesis or tumor progression, or serve as a source or target for therapeutic manipulations to improve the desirable outcome of melanoma. In addition, the structure of lead compounds produced endogenously or present in natural products could serve as a starting point for optimal drug design that would include targeting of specific receptors using tools of medicinal chemistry to develop optimal drugs or adjuvants for melanoma therapy. The large number of produced ligands, receptors, and signaling pathways associated with these secosteroids and Δ7 sterols/steroids, or pathways to be defined, present very complex problems for understanding their role in melanoma disease and for making rational choices in prevention or therapy. This also present a challenge for medicinal chemistry in developing or defining drugs with highly specific mechanisms of action. Such complex and diverse problems can be solved with the use of computational biology supported by AI. These would identify or confirm major regulatory pathways to be regulated by vitamin derivatives and their potential through in silico clinical trials run by the AI to define the most optimal strategy to be used in the clinic.

## Figures and Tables

**Figure 1 cancers-16-02262-f001:**
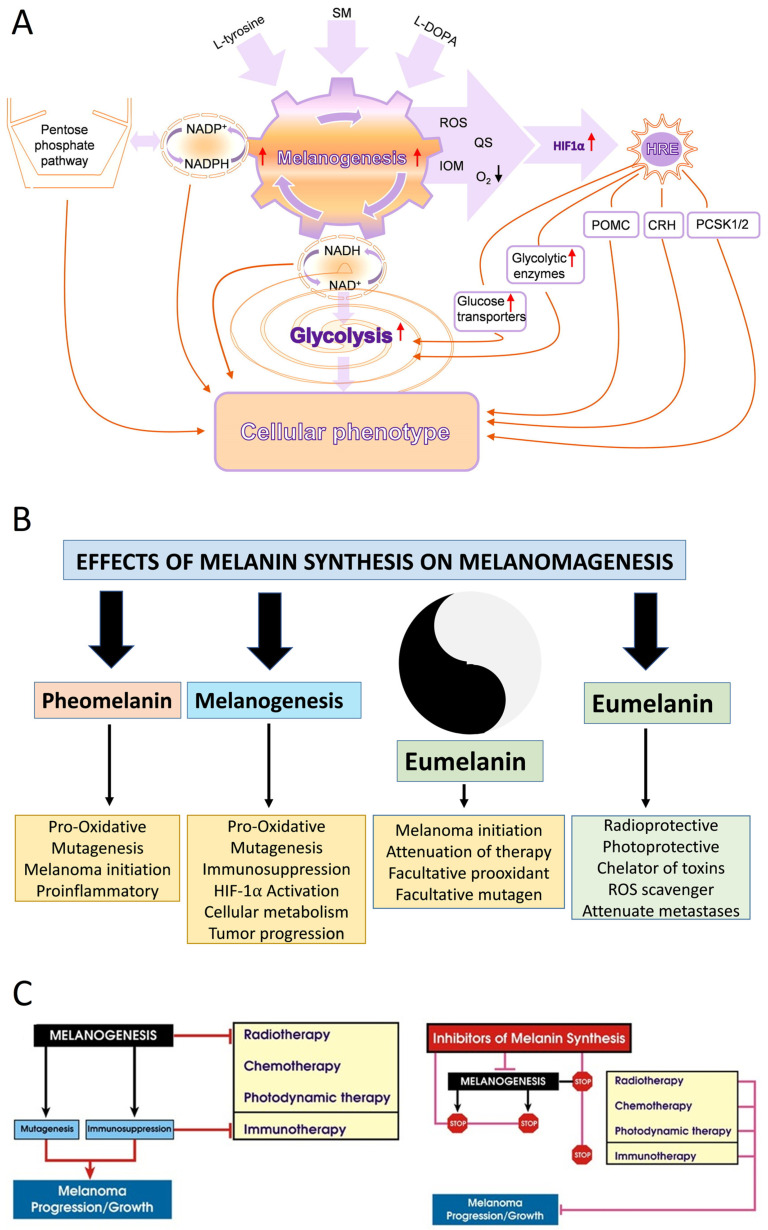
The effects of melanogenesis and melanin pigment on cellular metabolism, melanomagenesis and tumor progression and therapy. Panel (**A**) is reprinted from [236] and panels (**B**,**C**) are from [4] with permission from the publisher. Panel (**A**) shows how melanin pigment and melanogenesis regulate melanoma behavior. Panel (**B**) shows the effect of melanin pigmentation on melanomagenesis. Panel (**C**) outlines the effect of melanin and melanogenesis on therapy and disease outcome.

**Figure 2 cancers-16-02262-f002:**
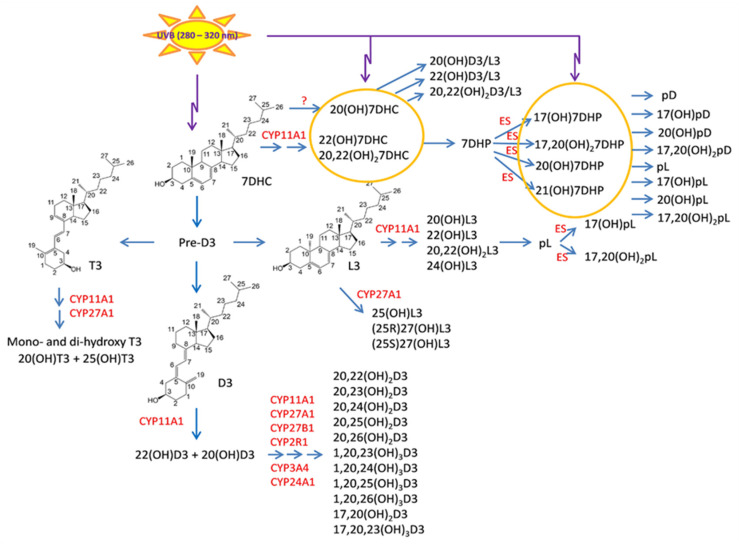
Novel metabolic pathways generating vitamin D3, 7DHC, lumisterol3, and tachysterol3 metabolites. Carbon numbers are numbered in parental structures. The number in abbreviations represent position of hydroxyl group, while the number after OH (hydroxyl group) represents umber of hydroxyl groups. D3 metabolism initiated by CYP11A1 generates at least 15 metabolites in cooperation with other CYP enzymes. Aside from hydroxylation of the side chain by either CYP11A1 or CYP27A1, metabolism of 7DHC and L3 by CYP11A1 also leads to the cleavage of side chain generating, respectively, 7-dehydropregnenolone (7DHP) and pregnalumisterol (pL) which can be further metabolized by steroidogenic enzymes (ES). The latter is in contrast to lack of such cleavage on the side chain of D3. However, 7DHP and its derivatives under UVB can transform to the corresponding pD, pL, and pT forms. The image taken with permission from the publisher [64], and modified from the original to include T3 and L3 activation by CYP27A1.

**Figure 3 cancers-16-02262-f003:**
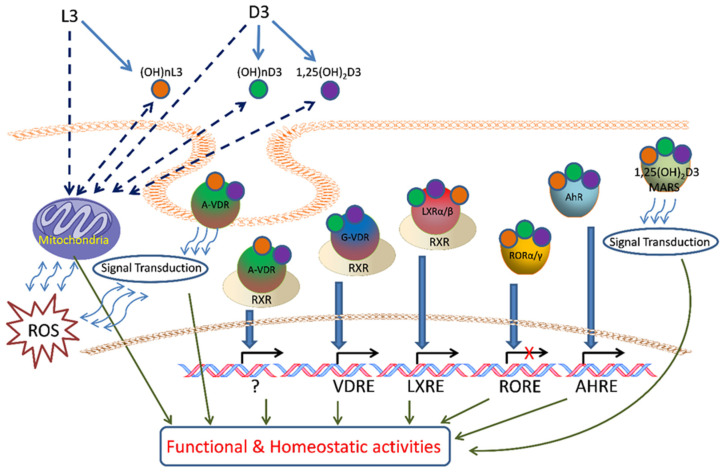
Proposed targets for vitamin D3 (D3) and lumisterol (L3) after metabolic activation. D3 and L3 are metabolized in mitochondria and microsomes to generate canonical 1,25(OH)2D3 and/or active metabolites (OH)nD3 and (OH)nL3. They act upon the A or G sites of VDR, RORs, LXRs, AhR, or 1,25D3 MARRS with activation of genomic or non-genomic signal transduction pathways to regulate cellular functional and homeostatic activities. Mitochondrial functions can also be affected by D3 and L3 activation or by the actions of their metabolites. Image taken from [64] with permission from the publisher.

**Figure 4 cancers-16-02262-f004:**
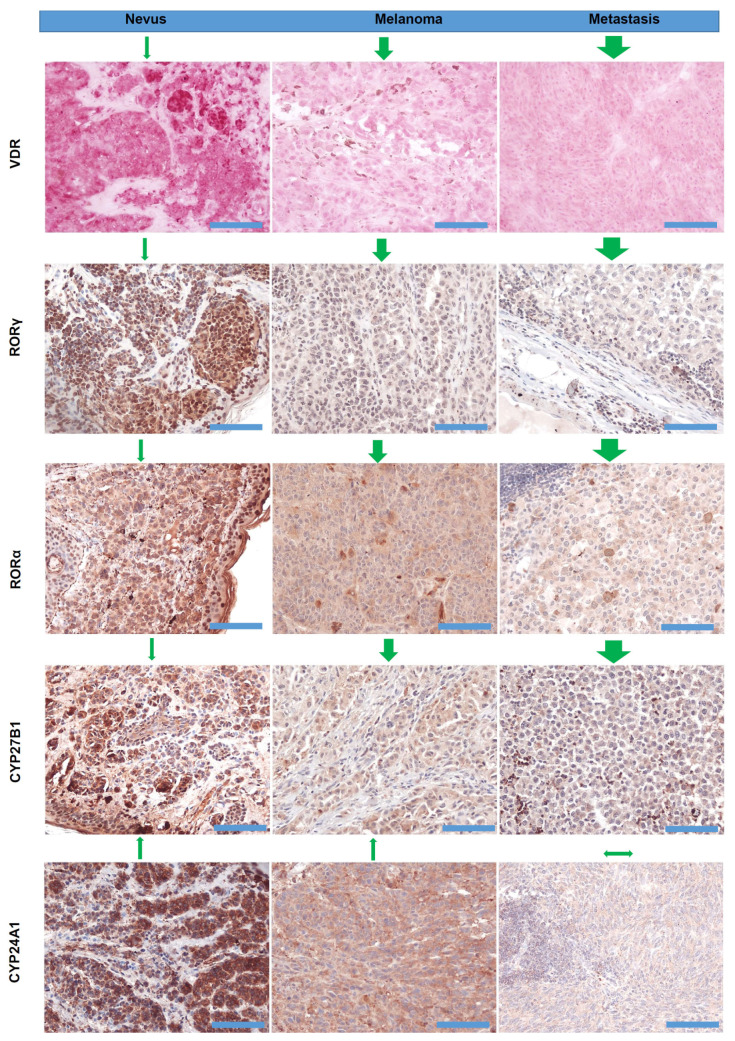
Representative images of VDR, RORα, RORγ, CYP27B1, and CYP24A1 immunostaining in nevi, melanomas, and metastases, as previously described [406,407,408,438]. VDR was detected with rat antibody (clone 9A7; Abcam, Cambridge, MA, USA; 1:75) and the positive cells were visualized with Red AP Substrate (Vector Laboratories, Burlingame, CA, USA). CYP27B1 was detected using rabbit antibody (clone H-90, Santa Cruz Biotechnology, Santa Cruz, CA, USA, 1:75), followed by visualization with ImmPACT NovaRED substrate (Vector Laboratories, Burlingame, CA, USA). RORα was detected with goat anti-RORα antibody (clone C-16, Santa Cruz Biotechnology, Dallas, TX, 1:25), and visualized with ImmPACT NovaRED (Vector Laboratories Inc., Burlingame, CA, USA). RORγ was detected with rabbit anti-RORγ antibody (generated and tested by Dr. Jetten’s lab, 1:50), followed by visualization with Vector NovaRED (Vector Laboratories Inc., Burlingame, CA, USA). CYP2A1 was detected with mouse antibody (Abcam, Cambridge, UK, 1:40) and the positive cells were visualized with ImmPACT NovaRED substrates (Vector Laboratories, Burlingame, CA, USA). Green arrows above the images reflect the direction and the scale of expression changes and in relation to the normal skin. Scale bars—100 µm.

**Figure 5 cancers-16-02262-f005:**
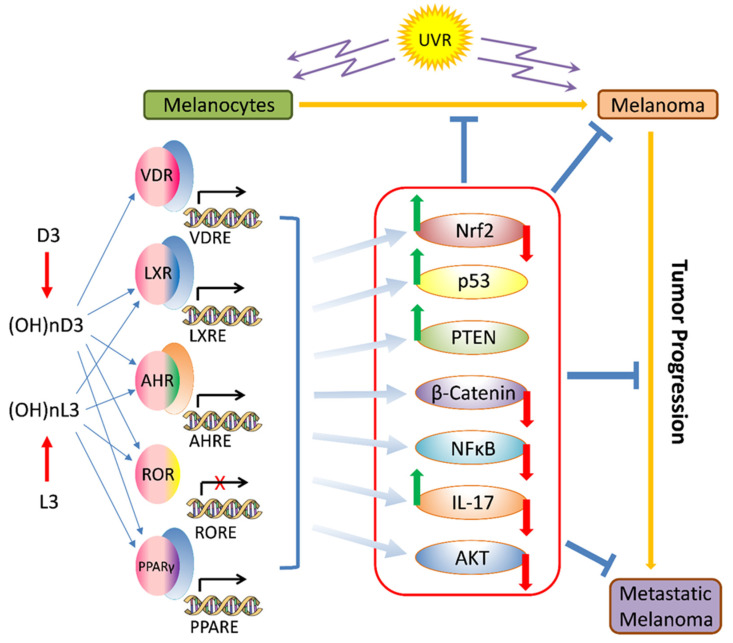
Potential role of vitamin D3 (D3), lumisterol (L3), and their active metabolites or synthetic derivatives on melanomagenesis and melanoma progression and therapy.

**Figure 6 cancers-16-02262-f006:**
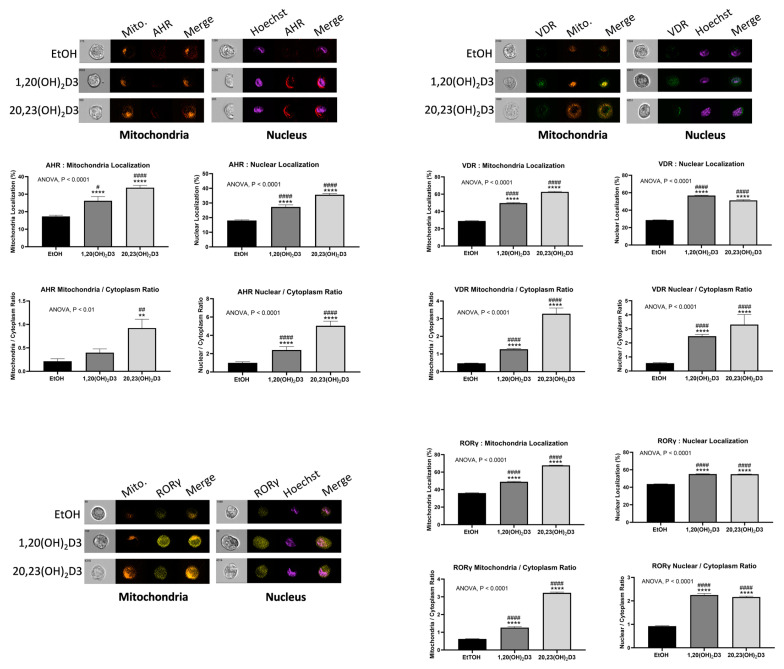
Nuclear and mitochondrial translocation of AHR, VDR, and RORγ by 1,20(OH)_2_D3 and 20,23(OH)_2_D3. Briefly, HaCaT cells, a human keratinocyte cell line [620,621], were incubated with 10^−7^ M 1,20(OH)_2_D3, 20,23(OH)_2_D3 or 0.1% ethanol (solvent control) for 6 h, fixed and stained with anti-AhR (Cat 565789 AF647, BD Biosciences), anti-VDR (sc-13133, AF 488, SantaCruz), or anti-RORγ antibodies (46-6981-82, PerCP-eFLUOR 710, Thermo Fischer) along with MitroTracker Orange (Thermo Fisher) for mitochondria localization or Hoechst dye (Thermo Fisher) for nuclear localization using protocols as described previously [355,465,488,492,624]. The stained cells were analyzed by imaging cytometry (Amnis Imagestream II). **Upper panel**: representative images of individual cells showing localization of AHR, VDR, or RORγ in cytoplasm, nucleus, or mitochondria following treatment with ethanol (control), 1,20(OH)_2_D3, or 20,23(OH)_2_D3, as indicated. The images show individual staining for AHR, VDR, or RORγ; Mito. for cells stained with MitoTracker orange; and Hoechst 33342 for staining of nucleus. Merged images show localization of AHR, VDR, or RORγ within mitochondria or nucleus. Bar graphs represent the quantitative analysis of AHR, VDR, or RORγ in mitochondria or nucleus as a percent (**mid panels**) or as a ratio relative to cytoplasmic levels. The data are from analysis of single cells images (n = 515 to 2339) acquired by imaging cytometry ± SE. One-way ANOVA (# *p* < 0.05; ## *p* < 0.01; #### *p* < 0.0001) or *t*-test (** *p* < 0.01; **** *p* < 0.0001) were used to determine statistical significance. The panel for AhR nuclear stain is reprinted from [488] with permission from the publisher.

**Figure 7 cancers-16-02262-f007:**
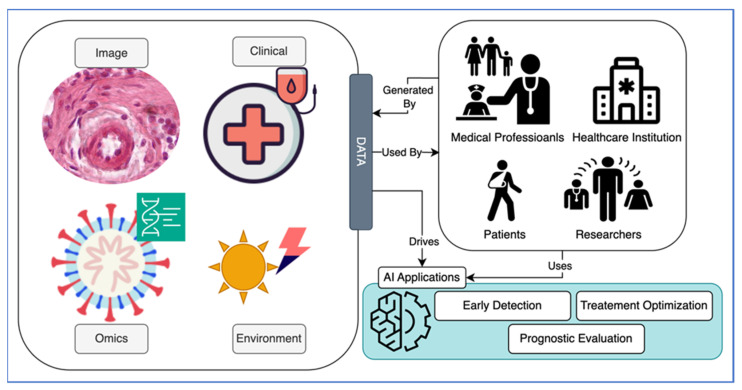
This diagram illustrates the integration of AI and data science in melanoma care, highlighting the progression from AI applications in general management to specific uses in detection, treatment, and prognosis, driven by diverse data sources, including genomic, imaging, clinical, and environmental information.

**Figure 8 cancers-16-02262-f008:**
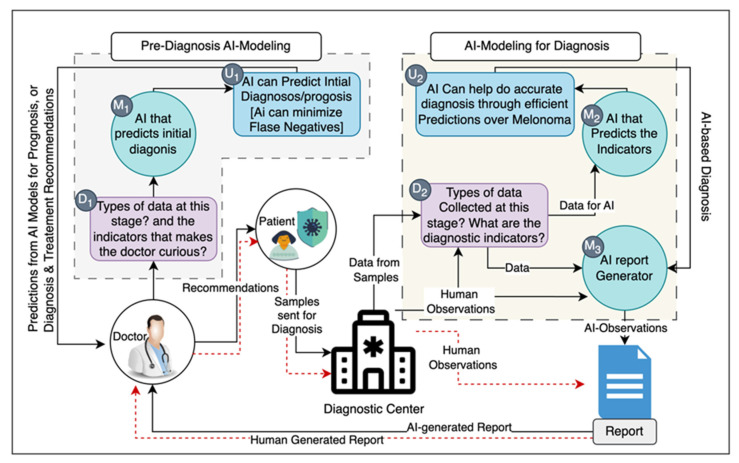
Harnessing AI for enhanced diagnosis and prognosis: a data-driven and time-efficient model.

## Data Availability

Not applicable.

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
