# Peer review of "Malignant Melanoma: An Overview, New Perspectives, and Vitamin D Signaling"

_cancers, 2024, doi:10.3390/cancers16122262_

Round 1
Reviewer 1 Report
Comments and Suggestions for Authors
This article offers a thorough examination of the impact of vitamin D and its derivatives on combating cancer, with a specific emphasis on melanoma. The authors skillfully underscore the significance of both classical and non-classical receptors, including VDR, RORα, and RORγ, in orchestrating the cellular and tissue responses essential for the anticancer properties of vitamin D. Additionally, the integration of AI in the discussion on bioinformatics considerations for melanoma diagnosis and treatment holds promise for further advancements in this field. Overall, the article provides a well-rounded exploration of key elements within this domain.
section 5.3.2, In in vivo studies, The text covers various findings regarding how vitamin D affects melanoma growth. However, it could offer more context to explain the importance of these results in the larger realm of melanoma research. Expanding on how these findings advance our knowledge of melanoma development and possible treatment approaches would enhance the discussion.
The text appropriately suggests the need for further studies to explore the effects of different vitamin D derivatives on melanoma growth and metastasis. However, it could elaborate more on specific research questions or hypotheses that could be addressed in future studies. This would help guide future research efforts in this area.
Section 5.5, discusses a range of data sources and repositories, including the Gene Expression Omnibus (GEO) and cBioportal. Offering specific instances or case studies illustrating the contributions of these data sources to melanoma research would enrich the reader's comprehension.
Author Response
Reviewer 1
This article offers a thorough examination of the impact of vitamin D and its derivatives on combating cancer, with a specific emphasis on melanoma. The authors skillfully underscore the significance of both classical and non-classical receptors, including VDR, RORα, and RORγ, in orchestrating the cellular and tissue responses essential for the anticancer properties of vitamin D. Additionally, the integration of AI in the discussion on bioinformatics considerations for melanoma diagnosis and treatment holds promise for further advancements in this field. Overall, the article provides a well-rounded exploration of key elements within this domain.
Reply.
We are pleased that our review was well received by both reviewers. We greatly appreciate the time and effort of reviewer 1 that allowed to improve our manuscript.
section 5.3.2, In in vivo studies, The text covers various findings regarding how vitamin D affects melanoma growth. However, it could offer more context to explain the importance of these results in the larger realm of melanoma research. Expanding on how these findings advance our knowledge of melanoma development and possible treatment approaches would enhance the discussion.
Reply
We appreciate the critique and the corresponding section 5.3.2 has been expanded as requested.
The text appropriately suggests the need for further studies to explore the effects of different vitamin D derivatives on melanoma growth and metastasis. However, it could elaborate more on specific research questions or hypotheses that could be addressed in future studies. This would help guide future research efforts in this area.
Reply
Following reviewer’s recommendations we expanded our discussion on questions and hypotheses that would guide future research efforts in this area, see revised manuscript.
Section 5.5, discusses a range of data sources and repositories, including the Gene Expression Omnibus (GEO) and cBioportal. Offering specific instances or case studies illustrating the contributions of these data sources to melanoma research would enrich the reader's comprehension.
Reply
We thank for the reviewer critique. The requested information is now added to the corresponding section 5.5.
Reviewer 2 Report
Comments and Suggestions for Authors
In the review, biochemical, cell-biological, research-relevant and clinically-relevant informations on vitamin D and the respective receptors in melanoma are presented. The review is very detailed and integrates the modern approach of AI technology.
In general, the figure legends of Figure 1 and 2 are too short. I.e. the meaning of colour markings (e.g. yellow circles; Fig.2), abbreviations (e.g. of molecule names) and the overall description is not sufficient to quickly understand the figures. In general a list of abbreviations would be nice.
Unfortunately, Figure 1 is not labelled A, B, C in the text and is generally poorly integrated into the running text. It would be nice to always have references in the text when it makes sense to take a closer look at the figure.
The reviewer used the list of lines to formulate questions that could possibly be elaborated on in the review.
Lane 176: Is there an explanation in the literature why BRAF and NRAS mutations are generated by intermittent sun exposure? Is this biochemically explainable?
Lane 199: Is there any evidence in the literature of a correlation between obesity and melanoma or not?
Lane 327: The aspect of vaccination therapy with mRNA agents is particularly innovative (mRNA vaccine; Moderna INC and Merck & Co Inc, 2025). It would therefore be nice if the study could be reported on in three or four sentences.
Lane 355/756: The point -melanoma itself can affect the host response at local and systemic levels through production of neurohormonal regulators- is also very interesting. The reviewer would be very pleased to receive a brief explanation of this.
Lane 849: The advice to look at Figure 4 seems completely out of place at this point. This is one of the main points that the images were poorly integrated.
Lane 883: A more detailed explanation of literature [54] is necessary. here.
Lane 1132: Figure 6 and the text belonging to Figure 6 does not fit into a review and the quality of the figure is also too poor.
Minor point: Is the sentence in lane 522 really complete?
Is it right to start the sentence in Lane 879 with –also- ?
Lane 890: -To- is twice in the text.
Comments on the Quality of English Language
The text should be checked for correct English. Some sentences sound strange (e.g lane 522, 890,. But I'm not a native speaker either.
Author Response
Reviewer 2
In the review, biochemical, cell-biological, research-relevant and clinically-relevant informations on vitamin D and the respective receptors in melanoma are presented. The review is very detailed and integrates the modern approach of AI technology.
Reply.
We are pleased that our review was well received by both reviewers. We greatly appreciate the time and effort of reviewer 1 that allowed to improve our manuscript.
In general, the figure legends of Figure 1 and 2 are too short. I.e. the meaning of colour markings (e.g. yellow circles; Fig.2), abbreviations (e.g. of molecule names) and the overall description is not sufficient to quickly understand the figures. In general a list of abbreviations would be nice.
Unfortunately, Figure 1 is not labelled A, B, C in the text and is generally poorly integrated into the running text. It would be nice to always have references in the text when it makes sense to take a closer look at the figure.
Reply
We greatly appreciate the reviewer critique. I compliance we provide revised and improved figure 1 and extended description in the figure legend to guide unprepared reader. This integrate better the figure with the text. The abbreviations are explained in the text when first used. We also provide additional explanation in figure 2 that carbon numbers are numbered in parental structures. The number in abbreviations represent position of hydroxyl group, while the number after OH (hydroxyl group) represents umber of hydroxyl groups. Other abbreviations are explained in the text.
The reviewer used the list of lines to formulate questions that could possibly be elaborated on in the review.
Lane 176: Is there an explanation in the literature why BRAF and NRAS mutations are generated by intermittent sun exposure? Is this biochemically explainable?
Reply
This has been explained on lines 182-192, as requested.
Lane 199: Is there any evidence in the literature of a correlation between obesity and melanoma or not?
Reply
This has been explained on lines 214-219, as requested.
Lane 327: The aspect of vaccination therapy with mRNA agents is particularly innovative (mRNA vaccine; Moderna INC and Merck & Co Inc, 2025). It would therefore be nice if the study could be reported on in three or four sentences.
Reply
This has been provided on lines 346-353, as requested.
Lane 355/756: The point -melanoma itself can affect the host response at local and systemic levels through production of neurohormonal regulators- is also very interesting. The reviewer would be very pleased to receive a brief explanation of this.
Reply
This has been now discussed on lines 384-391, as requested. Thank you for your request.
Lane 849: The advice to look at Figure 4 seems completely out of place at this point. This is one of the main points that the images were poorly integrated.
Lane 883: A more detailed explanation of literature [54] is necessary. here.
Reply
As requested the revised figure 4 with modified legend now is perfectly integrated into the manuscript flow. In addition the section In addition, section 5.2.1 has be edited to address the critique of detailed explanation.
Lane 1132: Figure 6 and the text belonging to Figure 6 does not fit into a review and the quality of the figure is also too poor.
Reply
Thank you for your critique. We are now providing revised high quality image to support the important points made in this section. There is a consensus among authors, some being leaders in corresponding fields, that this figure is a very important for listed effects of vitamin D analogs on mitochondrial functions. This is now also indicated in the revised corresponding subchapter.
Minor point: Is the sentence in lane 522 really complete?
Reply
It is complete and ends with citations.
Is it right to start the sentence in Lane 879 with –also- ?
Reply
Thank you for your attention, we have corrected this.
Lane 890: -To- is twice in the text.
Reply
Thank you for your attention to detail, we have corrected this.
Round 2
Reviewer 2 Report
Comments and Suggestions for Authors In my opinion, the review is acceptable for publication.Comments on the Quality of English Language
The text should be checked for correct English.